# Neuronal Yin Yang1 in the prefrontal cortex regulates transcriptional and behavioral responses to chronic stress in mice

Deborah Y. Kwon[1,2], Bing Xu[2,5], Peng Hu[1,2,5], Ying-Tao Zhao [1,2], Jonathan A. Beagan[2,3], Jonathan H. Nofziger[1,2], Yue Cui[1,2], Jennifer E. Phillips-Cremins [2,3], Julie A. Blendy[4], Hao Wu [1,2] & Zhaolan Zhou [1,2✉]

Although the synaptic alterations associated with the stress-related mood disorder major depression has been well-documented, the underlying transcriptional mechanisms remain poorly understood. Here, we perform complementary bulk nuclei- and single-nucleus transcriptome profiling and map locus-specific chromatin interactions in mouse neocortex to identify the cell type-specific transcriptional changes associated with stress-induced behavioral maladaptation. We find that cortical excitatory neurons, layer 2/3 neurons in particular, are vulnerable to chronic stress and acquire signatures of gene transcription and chromatin structure associated with reduced neuronal activity and expression of Yin Yang 1 (YY1). Selective ablation of YY1 in cortical excitatory neurons enhances stress sensitivity in both male and female mice and alters the expression of stress-associated genes following an abbreviated stress exposure. These findings demonstrate how chronic stress impacts transcription in cortical excitatory neurons and identify YY1 as a regulator of stress-induced maladaptive behavior in mice.

[1] Department of Genetics, University of Pennsylvania Perelman School of Medicine, Philadelphia, PA 19104, USA. [2] Epigenetics Institute, University of Pennsylvania, Philadelphia, PA 19104, USA. [3] Department of Bioengineering, University of Pennsylvania, Philadelphia, PA 19104, USA. [4] Department of Systems Pharmacology and Translational Therapeutics, University of Pennsylvania Perelman School of Medicine, Philadelphia, PA 19104, USA. [5]These authors contributed equally: Bing Xu, Peng Hu. ✉email: zhaolan@pennmedicine.upenn.edu

Major depressive disorder (MDD) is the leading global cause of disability, and its prevalence continues to rise[1]. Its effect on mental health is amplified by its high comorbidity with anxiety and other mental disorders. Further understanding of MDD's etiology is necessary in light of these concerns[1,2]. An abundance of epidemiological studies documenting MDD onset following adverse life experiences have led to its categorization as a stress-related illness[3,4]. Remarkably, rodents exposed to various forms of chronic stress also exhibit depressive- and anxiety-related phenotypes[5]. Female rodents show elevated endocrine and behavioral responses to stress compared to males[6,7], in consonance with the increased incidence of stress-related disorders in women[8–10]. These findings point to an evolutionarily conserved effect of stress on the brain and behavior, providing a basis for modeling the pathophysiology of stress-related mood and anxiety disorders in laboratory animals.

The prefrontal cortex (PFC) is the brain region responsible for top-down regulation of cognition, emotion, and behavior. Not only is the PFC acutely sensitive to trauma and stress exposure[11] but a concatenation of preclinical and clinical studies also support PFC dysfunction in MDD and other anxiety-related disorders[12,13]. A number of structural and functional changes in PFC pyramidal neurons—the primary glutamatergic excitatory cells in this brain region—have been reported in animals experiencing sustained stress or chronic glucocorticoid exposure[14–18], and decreased grey matter volume[19,20], synapse number[21], and altered glutamate levels[22,23] in the PFC have also been reported in MDD patients. These findings, taken together, have led to a glutamate hypothesis of depression, which theorizes that the disruption of glutamate-excitatory neurotransmission in the PFC leads to PFC hypoactivity, dysfunction, and impaired emotional regulation.

While the functional impact of chronic stress on the excitatory synapse is well studied, the molecular mechanisms that underlie or complement these synaptic alterations are poorly understood. Recent advancements in next-generation sequencing technology have enabled genome-wide transcriptional profiling in select brain regions of chronically stressed animals and MDD patients. These studies report broad alterations in whole-cell RNA populations across several brain regions associated with chronic stress and disease[7,24–26], suggesting that altered transcriptional programming may underlie the manifestation of stress-related mood disorders. While these findings inform us of the general effects of chronic stress on steady-state RNA levels, they cannot distinguish between alterations made in the nucleus during transcription or by post-transcriptional mechanisms in the cytoplasm. Moreover, the brain is a heterogeneous organ composed of numerous discrete cell types that are defined by unique transcriptomes. Chronic stress likely impacts transcription in a cell type-specific manner that cannot be fully resolved by profiling RNAs isolated from bulk tissues.

To probe the molecular mechanisms in the PFC that drive behavioral maladaptation to chronic stress, we performed complementary bulk and single-cell sequencing of nuclear RNA transcripts and mapped activity-dependent changes in chromatin architecture at a defined locus. This multipronged approach reveals key transcriptional regulators and gene-regulatory networks in neocortical excitatory neurons that are altered by chronic stress exposure. We also show that prolonged stress shapes these cells into a state of hypoactivity by reducing the transcription of synaptic genes involved in glutamatergic neurotransmission and restructuring genome architecture into a pattern associated with neuronal inactivity. We find that these alterations are mediated, in part, by a corticosterone-induced reduction of the transcriptional regulator, Yin Yang 1 (YY1), leading to the transcriptional misregulation of other transcription factors and compounding the effects of chronic stress on these

cells. Using adenoviral-mediated inactivation of *Yy1* in PFC excitatory neurons, we identify a cell type-specific role for YY1 as a regulator of adaptive transcriptional and behavioral responses to stress in male and female mice. Together, these findings provide insight into the molecular processes that transpire within the nuclei of neocortical excitatory neurons to drive maladaptive transcriptional and behavioral responses to chronic stress.

## Results

**Twelve days of chronic unpredictable stress drives depressive- and anxiety-like behaviors in adult male mice.** Considering the preponderance of epidemiological data linking chronic stress to major depressive and anxiety disorders, we first characterized the direct effects of chronic stress on behavior. We chose to use the chronic unpredictable stress (CUS) rodent model of depression due to its previously reported capacity to induce anhedonia and long-term depression-related phenotypes in adult male and female mice[27,28]. We expanded upon these findings by performing a battery of behavioral tests to assay depression- and anxiety-associated behaviors.

Our CUS paradigm consists of varying stressors delivered three times daily for variable lengths of time to prevent stress habituation (Supplementary Table 1). We subjected a cohort of adult male mice (9–10 weeks old) to twelve consecutive days of CUS followed by behavioral testing to assess the efficacy of the CUS paradigm. We first examined our mice for changes in body weight and food consumption, as stress exposure is known to affect these measures. Body weights of control and CUS animals were evenly distributed prior to CUS exposure (Fig. 1a). However, we found that twelve consecutive days of CUS produced significant changes in body weight between the two age-matched cohorts ($P < 0.0001$). While mice in the non-stressed control group gained ~1 g of weight on average over the twelve-day period, CUS prevented weight gain and produced weight loss in ~67% of the CUS cohort (Fig. 1b). We also found that CUS males consumed significantly less food than controls, even when food consumption was normalized to body weight to account for weight loss ($P < 0.01$; Fig. 1c).

We next assayed control and CUS mice for anhedonia, a hallmark symptom of depression, using the sucrose preference test. CUS males showed significantly decreased sucrose preference than controls ($P < 0.001$; 1d) and were also found to consume less total liquid over a 24-h period ($P < 0.05$; 1e). Loss of motivation is also associated with depression. Accordingly, we evaluated coat state, an indirect measure of grooming behavior, and nest-building to assess motivated behaviors in CUS-subjected animals[27]. We found that CUS mice exhibited deteriorated coat states, as reflected by significantly higher coat state scores ($P < 0.01$; Fig. 1f), as well as decreased nest scores after a 16-h overnight period with a fresh, intact cotton nestlet ($P < 0.05$; Fig. 1g). However, nests between control and CUS animals were virtually indistinguishable after 24 h, indicating that CUS induces a delay in nest-building and not physical impairment in nest-building ability. Lastly, we assayed behavioral despair, a depressive-like phenotype measured by the tail suspension test (TST), and observed a trend towards increased immobility in CUS mice compared to controls ($P = 0.07$; Fig. 1h).

Given the high comorbidity of anxiety and depression, we also assessed anxiety-related behaviors in CUS animals using the elevated zero maze (EZM) and the open field tests (OFT), both of which measure exploratory behavior related to anxiety. CUS mice were found to spend less time in the open arms of the elevated zero maze ($P < 0.01$; Fig. 1i), as well as reduced time in the center of the open field arena in the OFT ($P < 0.05$; Fig. 1j). Importantly, control and CUS mice traveled comparable distances in the OFT,

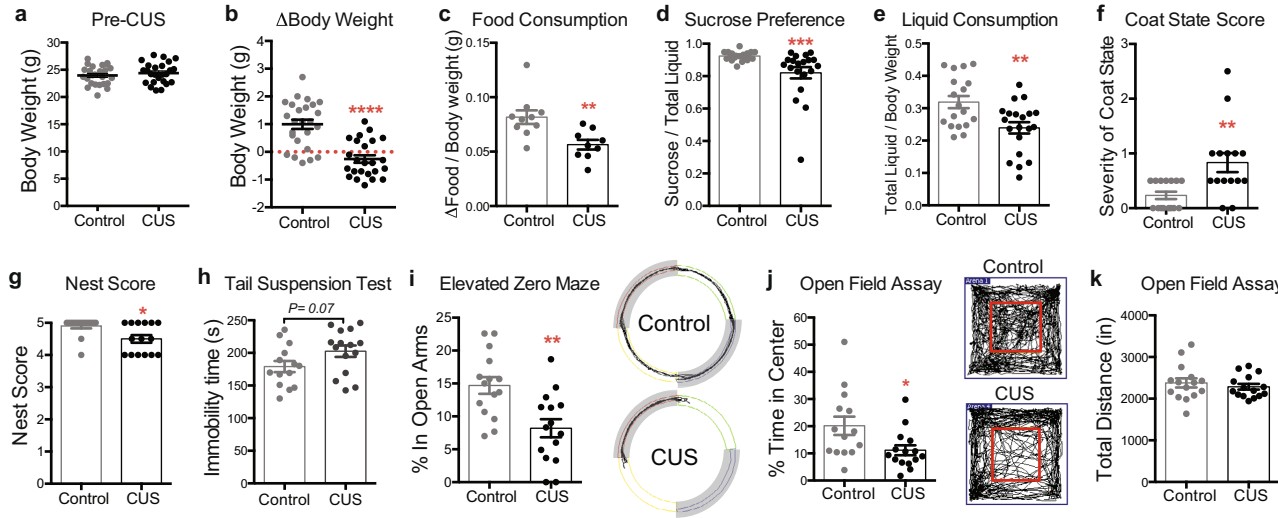

**Fig. 1 Twelve days of chronic unpredictable stress (CUS) drives a battery of depressive- and anxiety-like behaviors in adult male mice. a** Pre-CUS body weights of 9–10 weeks old control and CUS males ($n = 25$ per group). **b** CUS induces weight loss in adult males (Student's $t$-test; $P = 0.0001$; $n = 25$ per group; error bars, s.e.m.). **c** Food consumption of control and CUS males normalized to body weight (Mann–Whitney $U$-test; $P = 0.0021$; $n = 10$ controls, $n = 9$ CUS). **d** CUS decreases sucrose preference in male mice relative to controls (Mann–Whitney $U$-test; $P = 0.0004$; $n = 18$ controls, $n = 20$ CUS). **e** Liquid consumption of control and CUS males normalized to body weight (Mann–Whitney $U$-test; $P = 0.004$; $n = 18$ controls, $n = 20$ CUS). **f** CUS increases severity of coat scores relative to controls (Mann–Whitney $U$-test; $P = 0.002$; $n = 15$ per group). **g** CUS decreases nest scores relative to controls (Mann–Whitney $U$-test; $P = 0.01$; $n = 15$ per group). **h** CUS males show a trend toward enhanced immobility in the tail suspension test (Student's $t$-test; $P = 0.067$; $n = 14$ controls, $n = 15$ CUS). **i** CUS males show decreased percent time spent in the open arms of the elevated zero maze (Student's $t$-test; $P = 0.0017$; $n = 15$ per group). Representative tracks obtained from video-tracking software are shown on the right. Closed arms are shaded in grey. **j** CUS males show decreased percent time spent in the center of the open field arena (Student's $t$-test with Welch's correction; $P = 0.03$; $n = 14$ controls, $n = 15$ CUS) without **k** altered locomotor activity (Unpaired $t$-test; $P = 0.5$; $n = 14$ controls, $n = 15$ CUS). Representative tracks obtained from video-tracking software are shown in the middle, with the boundaries of the center of the arena demarcated in red. *$P < 0.05$; **$P < 0.01$; ***$P < 0.001$; ****$P < 0.0001$. Error bars represent s.e.m. and statistical tests were two-sided unless stated otherwise. Source data are provided as a Source data file.

indicating that the observed reduction in exploratory behavior was not due to CUS-induced differences in physical activity (Fig. 1k). Taken together, these findings demonstrate that twelve days of CUS produces robust depressive- and anxiogenic-like phenotypes in adult male mice.

**CUS deregulates transcription and alters chromatin folding in the prefrontal cortex.** We next sought to understand the transcriptional mechanisms underlying the observed effects of CUS on behavior. Gene expression changes can have a profound impact on neuronal physiology, and broad genome-wide alterations in gene expression have been reported in MDD and chronically stressed animals[7,24,25]. However, the vast majority of these RNA profiling experiments have assayed whole-cell RNA that provide an overview of gene expression changes but are unable to distinguish between those that arise during or post-transcription. Nuclear transcriptomes, which are mostly comprised of nascent pre-mRNA transcripts, provide a better representation of Pol II activity and chromatin state[29,30]. Thus, we performed RNA-sequencing (RNA-seq) on nuclei isolated from medial PFC (mPFC) tissues of control and CUS animals to determine how chronic stress impacts the transcriptional landscape of the cells that compose this stress-sensitive brain region.

We profiled nuclear RNA transcripts from 9 individual adult male mice (4 unstressed controls, 5 CUS). Our analysis uncovered 1362 differentially expressed transcripts in CUS samples (FDR < 0.05, Supplementary Data 1), 832 of which are downregulated and 530 upregulated (Fig. 2a; Supplementary Data 1). We found that the large majority of differentially expressed transcripts are composed of protein-coding genes (86.3%), but also identified differentially expressed long noncoding RNAs, including long

interspersed noncoding RNAs (lincRNAs, 7.1%), antisense transcripts (3.4%), and noncoding processed transcripts (1.5%), as well as small non-coding RNAs, which include microRNAs (1.3%) and small nucleolar RNAs (snoRNAs, 0.37), and 1 mitochondrial RNA (mtRNA) (Fig. 2b).

Gene ontology (GO) analysis of the differentially expressed protein-coding genes (DEGs) identified an enrichment of genes for neuron-associated proteins (Fig. 2d). These included genes encoding membrane proteins localized in the neuronal cell body, neuron projection, and somatodendritic compartment—indicating that a significant number of CUS-associated DEGs are found in neurons. GO analysis of the upregulated genes revealed an enrichment of receptor activity and cell signaling-related terms, suggesting that cells in the mPFC had been activated (Fig. 2e). We also found that downregulated genes, representing the majority of all CUS-associated DEGs (63.7%; Fig. 2c), primarily encode proteins related to DNA binding and Pol II-mediated transcription (Fig. 2f).

We observed that several of the downregulated genes encoding DNA binding proteins that were identified by GO analysis, including *Fos, Fosb*, and *Fosl2*, are regulated by neuronal activity. These data suggest that activity-regulated gene expression programs in the PFC might be altered by CUS exposure. We subsequently performed a pre-ranked gene-set enrichment analysis (GSEA[31]) to test for the overrepresentation of primary response genes, which are induced by neuronal stimulation[32], in our RNA-seq data. Indeed, we found a negative enrichment of primary response genes (normalized enrichment score = −2.05; FDR = 0.000) in our analysis, indicating that CUS leads to decreased neuronal activity in the PFC (Fig. 2g).

These findings led us to consider whether CUS induces a remodeling of higher-order genome architecture that reflect changes in neuronal activity. Genomes are organized into three-

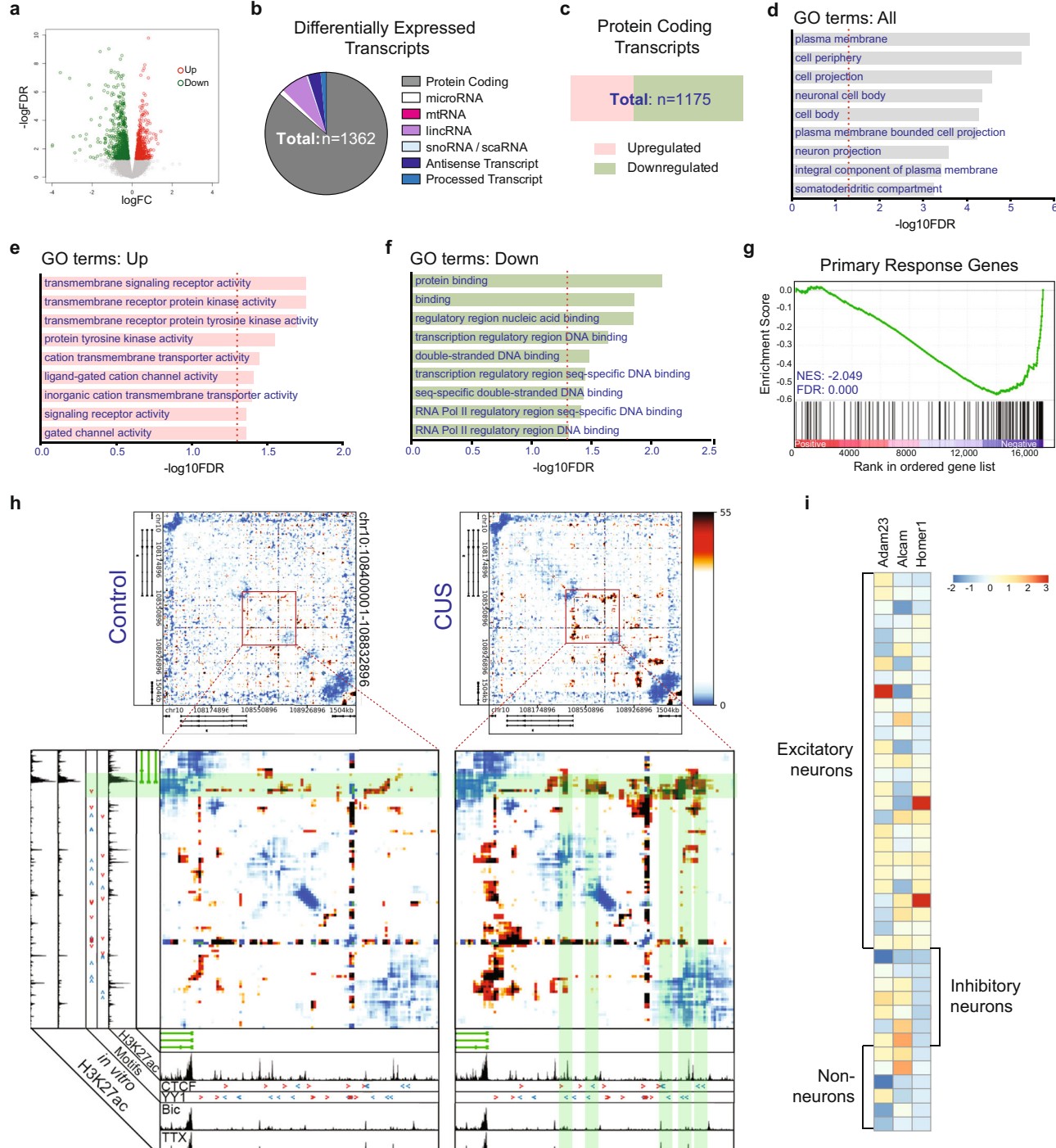

**Fig. 2 CUS impacts transcription and chromatin folding in the PFC. a** Significantly upregulated transcripts (red) and downregulated transcripts (green) from comparison of control and CUS PFC nuclear transcripts (FDR < 0.05). **b** Piechart exhibiting representation of differentially expressed RNA transcript populations. **c** Barchart portraying makeup of upregulated and downregulated differentially expressed protein-coding transcripts. **d**–**f** Top 9 most significantly enriched gene ontology (GO) terms for **d** all, **e** upregulated, and **f** downregulated CUS DEGs (FDR < 0.05). **g** GSEA plot charting negative enrichment of neuronal primary response genes in list of differentially expressed genes obtained from bulk nuclear RNA-seq analysis of control and CUS PFCs. NES, normalized enrichment score. **h** Background-corrected interaction frequency heatmaps displaying chromatin contacts in a 1.5-Mb region surrounding the *Syt1* gene in control and CUS frontal cortical tissues. The highlighted region in each heatmap marks the location of a zoomed-in plot. H3K27ac ChIP-seq track from adult cortical excitatory neurons control is shown below each heatmap. Enlarged heatmaps plot genomic locations and directionality of CTCF and YY1 binding sequences (red and blue arrowheads) in regions showing increased chromatin interactions in CUS mice (green shaded boxes). Legend depicts background-corrected interaction scores. **i** Heatmap showing expression of CUS DEGs, *Homer1*, *Adam23*, and *Alcam*, across the cortical cell populations defined by single-nucleus RNA-seq in the adult mouse neocortex. Legend depicts z-score of normalized gene expression.

dimensional structures, forming chromatin fibers that can be dynamically looped by proteins such as CTCF, cohesin, and YY1 to bring together regulatory genomic sequences with their targets[33–35]. We leveraged Chromosome-Conformation-Capture-Carbon-Copy (5C) sequencing data generated from mouse cortical neurons that were pharmacologically modulated with treatments of bicuculline (Bic) or tetrodotoxin (TTX), which stimulate and inhibit neuronal activity, respectively, to map activity-dynamic chromatin loops. We observed chromatin interactions between the gene encoding the pre-synaptic membrane protein, *Synaptotagmin-1* (*Syt1*), and upstream activity-decommissioned enhancers associated with H3K27ac that increased in TTX-inactivated neurons and decreased in Bic-stimulated neurons relative to untreated cells (Supplementary Fig 1a). These data demonstrate that chromatin architecture and histone acetylation around the *Syt1* locus is dynamically remodeled by neuronal activity. Notably, we found that chromatin contacts at these same regulatory elements also increased in frontal cortices of CUS-subjected mice compared to controls, indicating that chronic stress exposure restructures genome organization at this locus into a pattern associated with TTX-inhibition of neuronal activity (Fig. 2h, Supplementary Fig 1b). To identify the regulatory proteins accountable for these chromatin alterations, we also surveyed the genome upstream of *Syt1* for the presence of binding motifs of known architectural proteins—namely CTCF and YY1—which have known roles in organizing 3D chromatin structure. Using JASPAR, we uncovered YY1 and CTCF motif sequences in the regulatory regions showing increased chromatin interactions in CUS-subjected mice and TTX-treated cortical neurons (Fig. 2h, red and blue arrowheads).

Collectively, these results indicate that CUS exposure dynamically shapes the PFC into a state of neuronal inactivity by decreasing the transcription of neuronal activity-dependent genes and restructuring higher-order genome architecture into a pattern associated with synaptic silencing. Furthermore, our findings suggest that the zinc-finger transcription factors, CTCF and YY1, mediate chromatin folding at these loci in response to chronic stress and TTX administration.

**Broad distribution of DEG expression in the neocortex**. The cerebral cortex is a highly heterogeneous brain region composed of numerous neuronal and non-neuronal subtypes. Having determined that CUS alters transcription in cortical cells, we sought to construct cell type-specific models of transcriptional regulation by broadly classifying each DEG into a known cortical cell type. To accomplish this, we analyzed DEG expression across every cell type-specific cluster obtained in our previously published single-nucleus RNA-sequencing analysis of the mouse cortex[36]. However, we discovered that many CUS DEGs are expressed in multiple cortical cell types. While a subset of DEGs showed high expression in one cortical cell type, such as *Homer1*, which is primarily expressed in cortical excitatory neurons (Fig. 2i), many other DEGs displayed a ubiquitous or mixed pattern of expression in the cortex. *Adam23*, for example, encodes an extracellular matrix protein[37], that is expressed across many cortical clusters while *Alcam*, a cell adhesion molecule that has been found in blood-brain barrier and immune cells[38], not only shows expected expression in non-neuronal cells, but also inhibitory neurons and several subtypes of excitatory neurons (Fig. 2i). These expression patterns obfuscate the cell type origin of many CUS-associated DEGs and confound interpretation of cell type-specific CUS transcriptional networks.

**CUS-associated DEGs are cell type-specific**. To date, studies measuring stress-effects on gene expression have utilized bulk tissue samples, which provide a composite of gene expression changes. We had observed from our own nuclear transcriptome profiling that bulk tissue sequencing obscured the cellular origin of CUS-induced transcriptional alterations. This prompted us to leverage our single-nucleus droplet-based RNA-sequencing (sNucDrop-seq[36]) approach in the cerebral cortex to define cell type-specific transcriptional changes that occur in adult mice exposed to CUS.

Using quality filtering settings of >600 genes detected per nucleus, we retained 31,260 neocortical nuclei (12,402 uniquely mapped reads per nucleus) from adult control (15911 nuclei) and CUS (15349 nuclei) males, detecting, on average, 2566 transcripts per nucleus. Our analysis segregated nuclei into 26 distinct clusters, which we identified as excitatory (9 clusters), inhibitory (4 clusters), or non-neuronal (6 clusters) by their expression of known marker genes for major cortical cell types (Fig. 3a, b). Excitatory neurons (*Slc17a7*+) were further sub-categorized by their superficial-to-deep layer distribution within the cortex (layer 2 to layer 6), and every major sub-class of cortical inhibitory neurons (*Gad2*+) was also captured in our analysis (Fig. 3b). Non-neuronal clusters were identified as astrocytes (*Gja1*+), oligodendrocyte precursor cells (*Pdgfra*+), oligodendrocytes (Oligo1: *Opalin*+; Oligo2: *Enpp6*+), microglia (*Ctss*+), and endothelial cells (*Flt1*+). We also uncovered non-cortical contamination of our cerebral cortex dissections, such as striatal cells (~7%) and the connecting claustrum (~1%).

Because we had previously observed discrete clustering of cortical nuclei in seizure-induced mice[36], we asked whether we could identify CUS-dependent transcriptional states in the cortex. Upon comparing the segregation of cortical nuclei between control and CUS samples, we found that the representation and distribution of nuclei are largely similar under both conditions (Fig. 3c, d), indicating that CUS exposure does not lead to altered cellular composition of the neocortex.

Classifying cortical cell types allowed us to identify cell type-specific CUS-associated DEGs by comparing the expression of nuclear transcripts between control and CUS samples for each identified cluster (Supplementary Data 2). Our analysis uncovered CUS-associated DEGs across virtually every cortical cluster. We noted that excitatory neurons displayed the most transcriptional dysregulation, as measured by DEG number, with *Enpp2*+ layer 2/3 (L2/3_Enpp2) excitatory neurons showing the greatest number of CUS-associated DEGs (*n* = 606 DEGs; Fig. 3e). However, because L2/3_Enpp2 cells (and excitatory neurons in general) represented the largest cortical cell type captured by sNucDrop-seq, we asked whether the enlarged number of DEGs observed in these nuclei relative to other cortical cell types was due to increased statistical powering. To address this, we down-sampled 100 cells from each excitatory neuronal cluster as well as from 3 relatively large non-excitatory clusters, *PV*+ inhibitory neurons (Inh_Pv), astrocytes (Astro), and non-cortical dorsal striatal cells (STR). This approach enabled us compare an equal number of cells from each cluster in control and CUS mice. We then re-identified DEGs using the same statistical criteria and found that L2/3_Enpp2 neurons still contained the greatest number of DEGs and that other cortical excitatory neuronal subtypes, L5/6, L5, L6, L2/3_Ndst4, also showed more transcriptional deregulation than the non-excitatory neuronal groups analyzed (Supplementary Data 3). We also repeated this random down-sampling 30 times to ensure that this finding was not a biased result of the particular 100 cells selected, and found that L2/3_Enpp2 neuronal cluster still contained the highest proportion of DEGs (Supplementary Fig 2a). These findings indicate that the transcriptomes of cortical excitatory neurons, and L2/3_Enpp2 cells in particular, are preferentially impacted by chronic stress exposure.

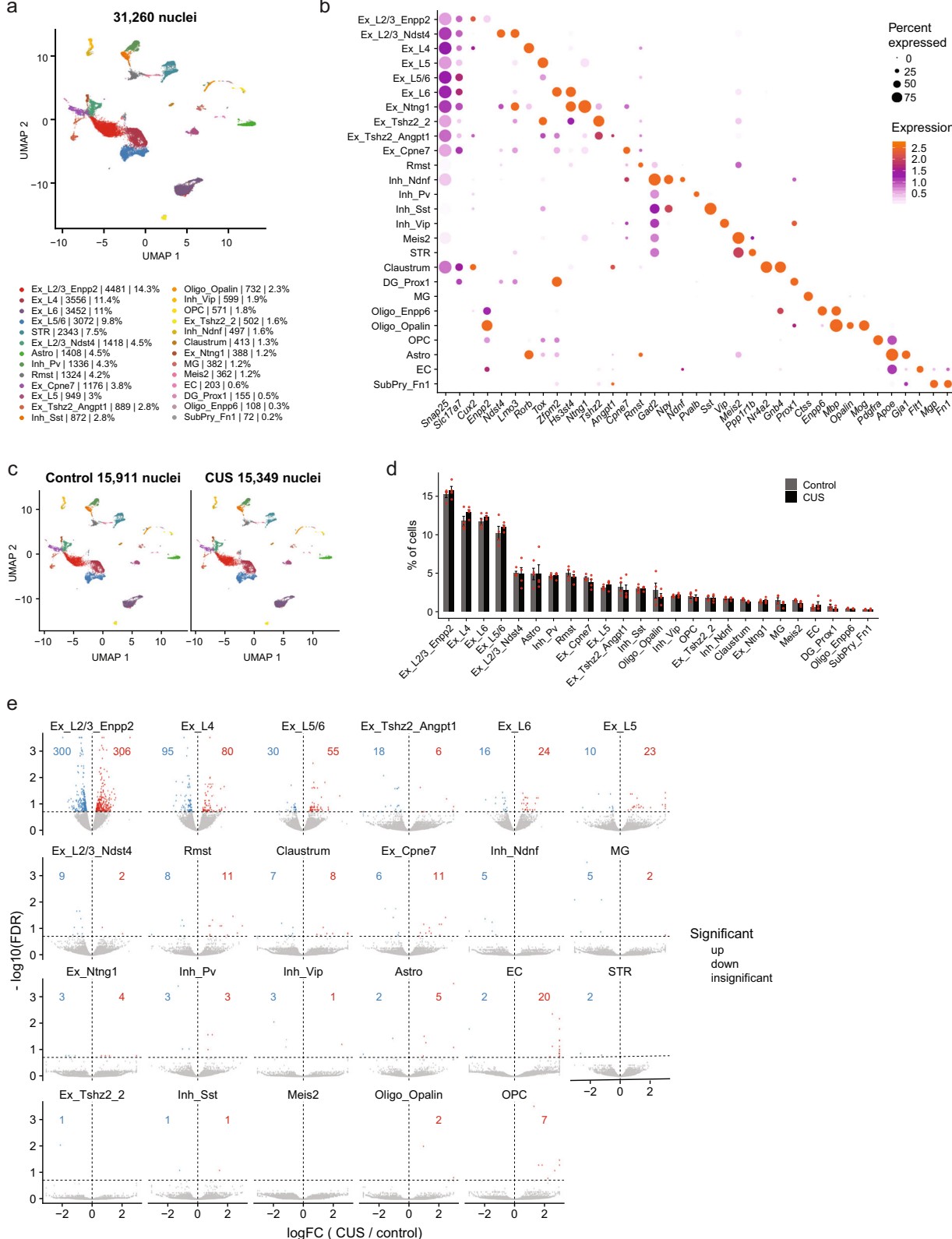

**Fig. 3 Identification of CUS-driven cortical cell type-specific gene expression changes using single-nucleus RNA-sequencing. a** Visualization of UMAP plot displaying 26 clusters segregated from all 31,260 nuclei isolated from adult control and CUS mouse cortices (*n* = 8 mice). Ex, excitatory neurons; Inh, inhibitory neurons; Astro, astrocytes; OPC, oligodendrocyte precursor cells; Oligo, oligodendrocytes; MG, microglia; EC, endothelial cells. **b** Bubble chart showing expression of cell type-specific marker genes for each cortical cell cluster. **c** UMAP plots depicting clusters identified from control (15,911 nuclei) and CUS nuclei (15,349 nuclei). **d** Percentage of control and CUS nuclei for every cortical cluster (*n* = 4 each group). **e** Volcano plots of differentially expressed genes between control and CUS cortical nuclei (FDR < 0.2). Upregulated genes are shown in red, downregulated genes are displayed in blue, and insignificant genes are shown in gray. Error bars represent s.e.m.

We also found a number of DEGs that are shared across discrete excitatory neuronal clusters; 36% of layer 4 DEGs (63/175 genes; Supplementary Fig 2b), 43.5% of layer 5/6 DEGs (37/85 genes; Supplementary Fig 2c), and 47.5% of layer 6 DEGs (19/40 genes; Supplementary Fig 2d) were shared with DEGs in L2/3. This partial overlap suggests that common molecular responses are induced across several different subclasses of cortical excitatory neurons, but that many CUS-associated transcriptional changes are unique to each neuronal subtype.

We then compared CUS-associated DEGs identified in the bulk nuclear RNA-seq and sNuc Drop-seq analyses in order to determine their cellular origin. We identified 66 individual genes that were classified as DEGs in the bulk nuclear RNA-seq dataset and also significantly dysregulated in and across several cortical excitatory, inhibitory, and non-neuronal cell populations obtained by sNucDrop-seq (Supplementary Fig 2e). A large majority of these shared DEGs showed a concordant pattern of gene deregulation (e.g., DEGs that were upregulated in the CUS bulk nuclear RNA-seq dataset were also upregulated by CUS in data generated by sNucDrop-seq), indicating that these DEGs are reproducibly regulated by CUS. Thus, application of these two methods of nuclear RNA-sequencing enabled discovery of transcriptional targets of CUS with cellular precision.

Considering the large number of DEGs obtained in the L2/3_Enpp2 cluster, we next examined the functional properties of these genes. We found that CUS-associated DEGs were significantly enriched in genes involved in glutamatergic neurotransmission and synapse structure, as indicated by GO enrichment analysis (Fig. 4a). Notably, an enrichment of excitatory synapse-related GO terms, pathways, and gene-sets were also identified in GWAS studies of MDD[39–41], suggesting the presence of shared transcriptional processes between CUS mice and MDD patients that impact glutamatergic synaptic function and lending further ethological validity to the CUS model of depression. Our sNucDrop-seq analysis identified several DEGs previously implicated in MDD, such as *Negr1*. *Negr1* has been identified in multiple GWAS studies of MDD but not found to be differentially expressed in bulk RNA-seq analyses of chronically stressed mouse cortices[24], including our own (Supplementary Data 2). Together, our data underscore the capability of single nucleus profiling in extracting subtle cell type-specific gene expression changes that are lost in whole tissue analysis.

Furthermore, GO analysis on L2/3 downregulated DEGs alone showed that the transcription of these synapse-related genes is decreased in response to CUS, in contrast to the ribosomal-related GO terms that were enriched in upregulated DEGs (Fig. 4b). The decreased synaptic gene expression in CUS excitatory neurons is consistent with previously reported observations of decreased dendritic spine density in PFC pyramidal neurons of stressed rodents and MDD subjects[21], as well as our own data demonstrating reduced neuronal activity in the PFCs of CUS mice (Fig. 2g). Together these data provide a potential mechanism for the known effects of chronic stress on synaptic volume and activity in cortical excitatory neurons.

**CUS decreases YY1 regulatory activity in layer 2/3 excitatory neurons**. Gene expression is a precisely orchestrated and cell type-specific process—regulated, in part, by networks of transcription factors (TFs) and co-factors within the cell. To gain mechanistic insight into the deregulation of transcription observed in CUS L2/3_Enpp2 nuclei, we sought to identify specific gene regulatory networks (GRNs) that are altered by CUS in this excitatory neuronal subtype. Using SCENIC[42], which infers enriched GRNs associated with specific transcription factors from single-cell RNA-seq data, we detected 346 TFs in L2/3_Enpp2 cortical nuclei whose binding motifs are significantly enriched in their co-regulated GRNs. Among these TFs, 41 showed differential GRN activity in CUS nuclei, and 5 of these themselves were significantly dysregulated in expression in our sNucDrop-seq data (Fig. 4c, Supplementary Fig 3a, b). We also identified altered regulatory activity of several TFs previously implicated in chronic stress, including FOS, FOSB, and CREM, which are significantly upregulated in CUS L2/3_Enpp2 neurons compared to controls (Fig. 4c, Supplementary Fig 3a).

Interestingly, we found that the regulatory activity of CTCF and YY1 in L2/3_Enpp2 nuclei was significantly altered by CUS in opposing directions (Fig. 4c, Supplementary Fig 3a,b). This finding is consistent with CTCF and YY1's preference for different regulatory elements; while CTCF tends to occupy insulator elements, YY1 has been shown to preferentially occupy active enhancers and promoters[43]. These proteins are of particular interest given CTCF and YY1's role in regulating cell type-specific gene expression programs and our own data showing CTCF and YY1 motifs at regulatory regions of dynamic chromatin looping in CUS frontal cortices (Fig. 2i). These findings suggest that CUS decreases YY1 binding and subsequently increases CTCF-mediated chromatin interactions at this genomic region.

Moreover, a recent study reported that a number of SNPs associated with MDD disrupt the binding motifs of these chromatin regulators[44], indicating that perturbation of genome architecture by CTCF and YY1 may be a functional mechanism underlying MDD etiology. While the nuclear transcript levels of *Ctcf* itself were insignificantly altered between control and CUS L2/3_Enpp2 neurons, we found that *Yy1* was significantly decreased by CUS in this cell population (Supplementary Data 2). This finding was of great interest given that *Yy1* had been previously identified as a hub gene in a whole blood gene expression network analysis of MDD, and its expression was found to be negatively correlated with MDD status[45]. Further examination revealed that the decreased expression level of *Yy1* (Supplementary Fig 4a) was largely driven by a significant decrease in the percentage of L2/3_Enpp2 nuclei expressing *Yy1* transcripts in CUS conditions (Fig. 4d, Supplementary Fig 4b). Furthermore, this CUS-associated decrease in *Yy1*-expressing L2/3_Enpp2 neurons captured by our single-nucleus RNA-seq was observed in both batches of sequencing that we performed, indicating that this finding is reproducible across cohorts (Supplementary Fig 4b). These data collectively demonstrate that CUS decreases YY1 regulatory activity, in part, through downregulating *Yy1* transcription.

Despite its constitutive expression in the brain, studies of YY1 have been generally constrained to its role in early CNS development and its molecular function in the adult brain is largely unexplored. To better understand YY1's regulatory function in adult cortical excitatory neurons, we performed a GO enrichment analysis on the 241 genes that comprise the YY1 GRN/regulon as determined by SCENIC (Supplementary Data 2). We found that YY1-regulated genes primarily encode nuclear proteins that regulate RNA Pol II-mediated transcription (Fig. 4e), consistent with the enrichment of RNA Pol II-mediated transcription GO terms seen in DEGs downregulated in our CUS bulk nuclear transcriptomic analysis (Fig. 2f). These data suggest that decreased YY1 activity may, in part, regulate the transcriptional downregulation observed in our bulk nuclear transcriptome analysis.

In accordance with YY1's known ubiquitous expression, we detected *Yy1* transcripts in every major cortical cell type in our sNucDrop-seq dataset (Supplementary Fig 4c). We then compared the percentage of *Yy1*-expressing nuclei between control

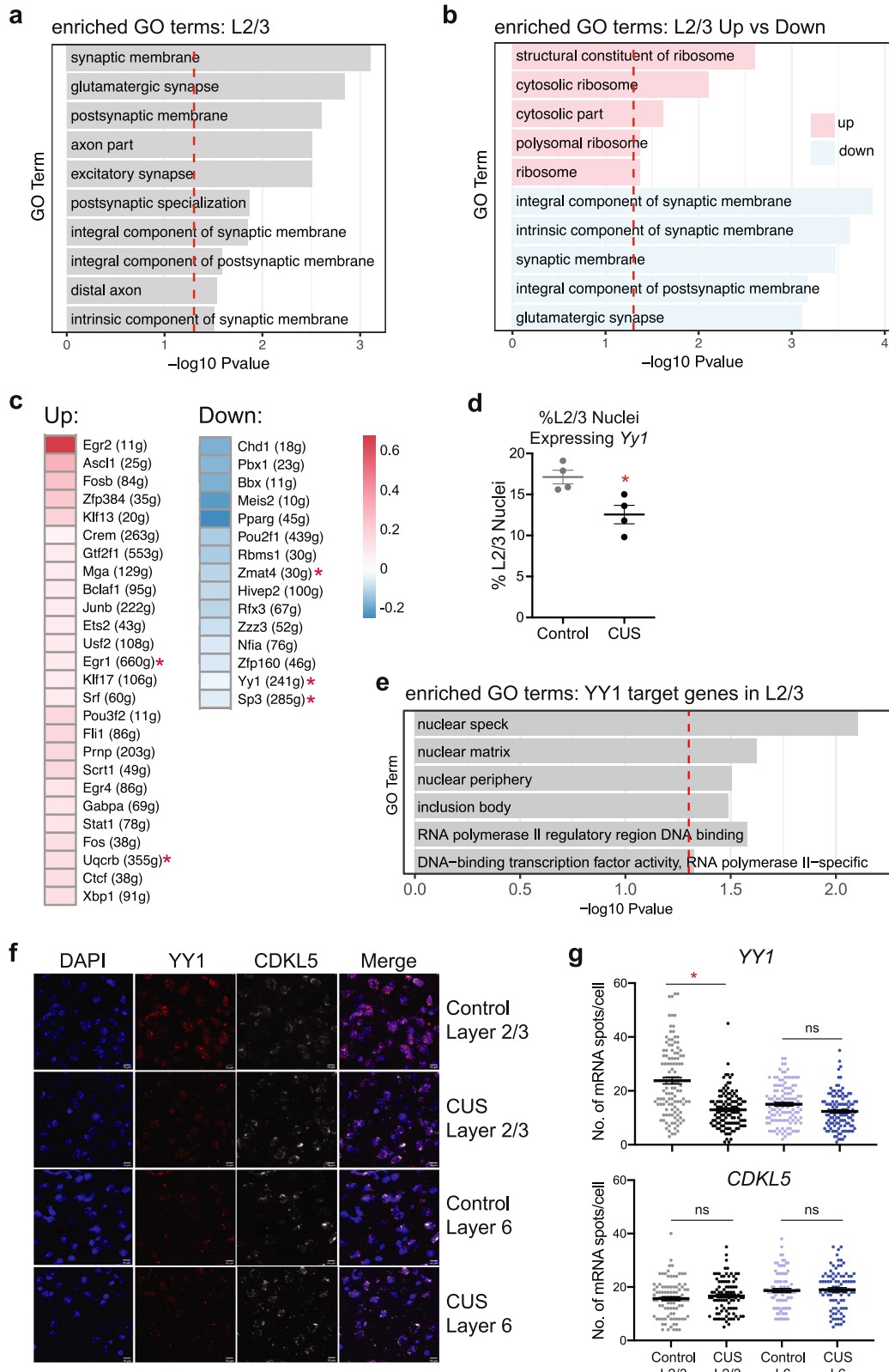

and CUS samples for every major subtype of excitatory and inhibitory neurons. We not only discovered that the percentage of *Yy1*-expressing nuclei decreased in CUS-isolated L2/3_Enpp2 neurons relative to controls, but that it also trended toward a decrease in nearly every CUS-associated excitatory neuronal subtype (Supplementary Fig 4d; $P < 0.1$). Remarkably, this reduction in *Yy1* transcripts was not observed in any inhibitory

neuronal subtype (Supplementary Fig 4d), suggesting an excitatory neuron-specific effect of CUS on *Yy1* expression.

Having observed the largest effect of CUS on *Yy1* transcripts in layer 2/3 cortical neurons by sNucDrop-seq, we next sought to validate this finding by an orthogonal method. Thus, we performed RNAscope on coronal sections of brains extracted from control and CUS-subjected mice, using probes designed to

**Fig. 4 Synaptic gene expression and YY1 regulatory activity are decreased in CUS L2/3 excitatory neurons. a** L2/3 DEGs are enriched for synapse- and axonal-related gene ontology (GO) terms (FDR < 0.5). **b** Upregulated L2/3 DEGs (pink) are enriched for ribosomal-related GO terms. Downregulated L2/3 DEGs (blue) primarily encode synapse-related proteins. **c** Heatmap of CUS L2/3 SCENIC results showing 26 transcription factors that are significantly upregulated in activity and 15 that are downregulated in activity relative to controls. Legend depicts normalized mean AUC values for CUS nuclei relative to control nuclei. **d** Percentage of L2/3 control and CUS nuclei expressing *Yy1* (Unpaired *t*-test, $P = 0.017$; $n = 4$ per group). **e** YY1 target genes in L2/3 neurons encode nuclear proteins and RNA Pol II-mediated transcription-related GO terms. **f** Representative RNAscope images for *Yy1* and *Cdkl5* in medial PFC tissues taken from control and CUS mice. **g** Quantification of *Yy1* and *Cdkl5* mRNA puncta number per cell in layer 2/3 and layer 6 are shown on the right (Linear mixed-effect model; $P = 0.02$; $n = 30$ cells per animal, $n = 3$ animals per treatment). *$P < 0.05$. Error bars represent s.e.m. Source data are provided as a Source data file.

detect *Yy1* mRNA. We also performed RNAscope for *Cdkl5*, whose expression is unaltered by CUS, in the same tissue sections as a negative control. We then compared the numbers of *Yy1* and *Cdkl5* mRNA puncta in layer 2/3 cells of the medial PFC (mPFC) to those captured in layer 6. In agreement with our sNucDrop-seq results, we observed a modest and insignificant ~20% reduction in the number of Yy1 mRNA puncta in mPFC layer 6 ($P = 0.276$) and a greater ~44% reduction of *Yy1* mRNA puncta in layer 2/3 cells that reached statistical significance (Fig. 4f, g; $P = 0.0198$). This finding was in contrast to *Cdkl5*, which was neither altered in layer 2/3 nor in layer 6 of the mPFC (Fig. 4f, g; L2/3, $P = 0.466$; L6, $P = 0.833$), supporting our sNucDrop-seq finding.

Taken together, these findings indicate a potential function for YY1 in mediating the transcriptional consequences of chronic stress in cortical excitatory neurons and in layer 2/3 pyramidal neurons, in particular.

**Chronic CORT administration downregulates *Yy1* gene expression and decreases YY1 protein levels.** Stress exposure alters multiple hormone signaling pathways in the brain. These include signaling mechanisms mediated by glucocorticoids, which are secreted upon activation of the hypothalamic-pituitary-adrenal axis and regulate downstream gene expression. To determine whether CUS decreases YY1 through glucocorticoids, we used primary cultures of mouse cortical neurons to assay the effect of stress-released corticosterone (CORT)—the principal rodent glucocorticoid—on YY1 in a relatively homogenous population of cells comprised mostly of excitatory neurons. In addition to YY1, we also examined the expression of *Nuclear receptor subfamily 3 group C member 1* (*Nr3c1*), which encodes the glucocorticoid receptor (GR). We used *Nr3c1* expression as a positive indicator of CORT exposure in this experimental model, given that *Nr3c1* transcription is auto-regulated by CORT-controlled feedback mechanisms[46] and reportedly mediated by YY1 activity[47]. In agreement with these prior findings, we also found that *Nr3c1* is a gene member of the YY1 GRN/regulon (Supplementary Data 2).

We treated cortical neuron cultures with a physiologically relevant concentration of CORT (1 μM[48]) for varying lengths of time (0 h, 3 h, 24 h, 72 h, 1.5 wk) followed by real-time PCR (RT-PCR) and western blot analyses (Supplementary Fig 5a). We found that treating primary cortical neurons with 1 μM CORT for 72 h and 1.5 wk significantly decreased nuclear expression of *Yy1* (Supplementary Fig 5c), and *Nr3c1* (Supplementary Fig 5b) relative to vehicle treated (0 h) cells. In contrast, 3 hr and 24 h exposure to CORT had no effect on *Yy1* transcript levels (Supplementary Fig 5c). Importantly, we found that 1.5 weeks of CORT treatment significantly decreased YY1 steady-state protein levels as well (Supplementary Fig 5d), indicating that chronic CORT administration, but not acute exposure, decreases YY1 at both the transcript and protein levels. Together with our in vivo sNucDrop-seq data, these findings indicate that CUS exposure likely decreases YY1 activity in cortical excitatory neurons through prolonged production of CORT.

**In vivo downregulation of YY1 in PFC excitatory neurons increases stress susceptibility in mice.** The downregulation in YY1 expression and activity that we observed in CUS cortical excitatory neurons indicated that YY1 might mediate the transcriptional and behavioral effects of stress. Thus, we next investigated the functional role of cortical YY1 in vivo. Due to YY1's essential role in early cortical development and its ubiquitous expression, we adapted a genetic strategy to selectively ablate YY1 expression in PFC excitatory neurons of adult male mice. We performed bilateral PFC injections in 9–12 week old male mice carrying a floxed *Yy1* allele[49] (*Yy1^{fl/fl}*) with adeno-associated virus (AAV) expressing either *CamKII* promoter-driven eGFP (hereafter referred to as YY1-exGFP) or Cre recombinase fused to eGFP (subsequently referred to as YY1-exKO). This approach restricts viral expression of eGFP/eGFP-Cre to *CamKII*+ excitatory neurons in the PFC (Fig. 5b). AAV-infected PFC tissues from Yy1-exKO mice showed a ~50% reduction in *Yy1* gene expression compared to YY1-exGFP controls by RT-PCR (Fig. 5c) as well as a ~50% decrease in YY1 protein levels (Fig. 5d). These results are consistent with previous studies that have shown a ~50% reduction in CTCF, another ubiquitously expressed transcriptional regulator, in cortical and hippocampal tissues of CTCF floxed mice expressing *CamKII-Cre*[50–52]. These results demonstrate the high cellular heterogeneity of the cortex and underscore the need for studies that analyze cell type-specific functions of ubiquitously expressed factors such as YY1.

To investigate the behavioral consequences of deleting *Yy1* in PFC excitatory neurons, we performed the sucrose preference, nesting, and open field assays in both cohorts of mice following 3 weeks of recovery (Supplementary Fig 6a). We found that in vivo deletion of *Yy1* in PFC excitatory neurons alone did not induce a robust depressive- and anxiety-like state in mice, as determined by the comparable sucrose preference (Supplementary Fig 6b) and open field responses (Supplementary Fig 6d, e) between YY1-KO males and YY1-exGFP controls. YY1-KO mice, however, did exhibit a trend toward decreased nesting behavior (Supplementary Fig 6c; $P = 0.08$). These results indicate that loss of YY1 in PFC excitatory neurons alone does not significantly perturb PFC function to drive complex behaviors under these experimental conditions.

Given that our experimental and computational analyses uncovered chronic stress-associated decreases in *Yy1* transcription and regulatory activity in neocortical excitatory neurons, we reasoned that decreased YY1 function in this neuronal population might influence stress coping in vivo. Thus, to determine whether selective loss of YY1 in adult PFC excitatory neurons influences stress sensitivity, YY1-exGFP and YY1-exKO animals were subjected to an abbreviated form of the CUS paradigm that consists of 3 days of CUS stressors (subsequently referred to as aCUS; Fig. 5a, Supplementary Table 2) before undergoing the sucrose preference, nesting, open field, and tail suspension tests. Notably, we found that loss of YY1 in this cell population significantly enhanced stress susceptibility in male mice. YY1-exKO males spent less time in the center of the open field arena

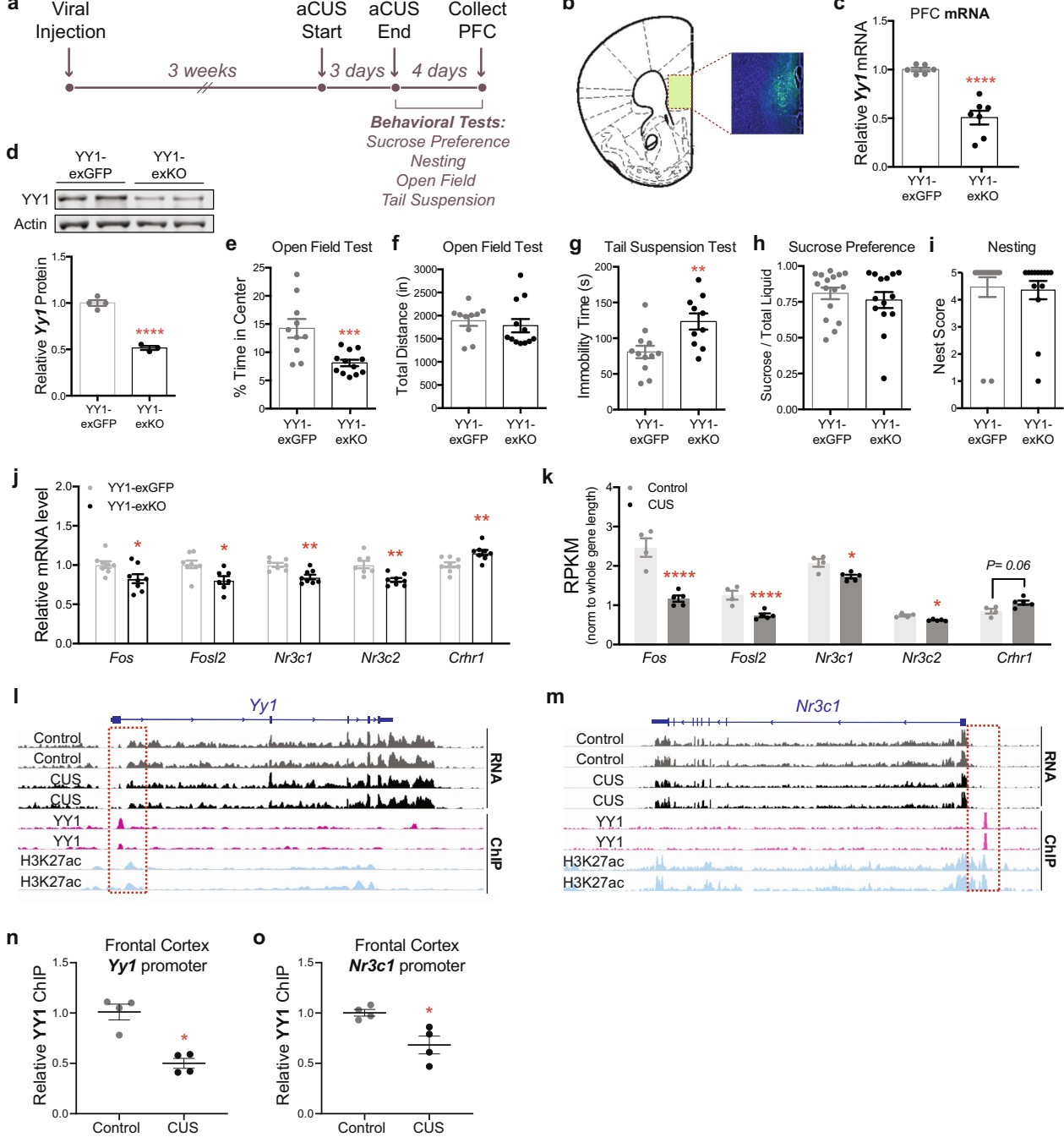

compared to aCUS-exposed Yy1-exGFP controls (Fig. 5e) with no alteration in locomotor activity (Fig. 5f). YY1 inactivation in PFC excitatory neurons also significantly increased immobility time in the tail suspension test in aCUS mice (Fig. 5g), but did not impact sucrose preference or nesting behavior compared to aCUS YY1-exGFP males (Fig. 5h, i).

Having discovered that Yy1-exKO mice subjected to aCUS develop CUS-associated phenotypes, we hypothesized that their behavior was driven by a pattern of neuronal gene deregulation comparable to CUS-exposed males. First, we compared the expression of the genes encoding the stress hormone receptors, Nr3c1 and Nr3c2, and corticotropin-related hormone (CRH) receptor, Crhr1. These genes have been extensively implicated in stress-related mood and anxiety disorders and in chronically stressed rodents. Gene expression assays performed in medial

PFC tissues from Yy1-exKO males subjected to aCUS showed a significant reduction in both Nr3c1 and Nr3c2 and increased Crhr1 expression relative to aCUS-stressed Yy1-exGFP controls (Fig. 5j). We compared these gene expression changes to those obtained in bulk nuclear RNA-seq experiments from CUS-subjected mice. In agreement with data obtained from aCUS Yy1-exKO mice, RNA-seq analysis of CUS PFC tissues showed a significant reduction in both Nr3c1 and Nr3c2 expression and a trend toward increased Crhr1 expression ($P = 0.06$) compared to non-stressed control males (Fig. 5k).

The stress-responsive neuronal activity genes, Fos and Fosl2, are components of the activating protein-1 (AP-1) transcriptional complex. Previously published work documenting reduced expression of Fos and Fosl2 in the PFCs of chronically stressed rodents[24,53,54] and our own data showing dysregulated FOS gene

**Fig. 5 Selective genetic deletion of *Yy1* in PFC excitatory neurons enhances stress vulnerability in adult male mice. a** Timeline of AAV injections, aCUS, and behavioral experiments. **b** Representative image of GFP expression in medial PFC of AAV-injected mouse. **c** Quantification of *Yy1* mRNA levels in YY1-exKO mice relative to YY1-exGFP mice (Unpaired *t*-test with Welch's correction; $P < 0.001$; $n = 6$ YY1-exGFP, $n = 7$ YY1-exKO). **d** Representative western blot showing YY1 and β-actin proteins in medial PFC tissue lysates from YY1-exGFP and YY1-exKO mice. Semi-quantification of YY1 knockdown (normalized to β-actin) is shown on the right (Unpaired *t*-test; $P < 0.001$; $n = 4$ YY1-exGFP, $n = 3$ YY1-exKO). **e** Decreased exploratory behavior in the open field arena in aCUS-exposed mice harboring selective loss of YY1 in PFC excitatory neurons (Unpaired *t*-test with Welch's correction; $P = 0.001$; $n = 10$ YY1-exGFP, $n = 12$ YY1-exKO). **f** Total locomotion in the open field test is unaltered (Unpaired *t*-test; $P = 0.58$; $n = 10$ YY1-exGFP, $n = 12$ YY1-exKO). **g** Loss of YY1 in PFC excitatory neurons increases immobility in aCUS-exposed male mice (Unpaired *t*-test; $P = 0.007$; $n = 12$ YY1-exGFP, n=10 YY1-exKO). **h, i** Yy1-exKO males subjected to aCUS show no alterations in behavior during the **h** sucrose preference (Mann–Whitney *U*-test; $P = 0.76$; $n = 16$ Yy1-exGFP, $n = 14$ Yy1-exKO) and **i** nesting (Mann–Whitney *U*-test; $P = 0.31$; $n = 16$ Yy1-exGFP, n=14 Yy1-exKO) assays. **j** Quantitative RT-PCR measurements of *Fos*, *Fosl2*, *Nr3c1*, *Nr3c2*, and *Crhr1* mRNA levels in aCUS YY1-exKO mice relative aCUS YY1-exGFP controls (Unpaired *t*-test; *Fos* $P = 0.02$; *Fosl2* $P = 0.01$; *Nr3c1* $P = 0.0023$; *Nr3c2* $P = 0.0024$; *Crhr1* $P = 0.006$; $n = 8$). **k** RPKM values for *Fos*, *Fosl2*, *Nr3c1*, *Nr3c2*, and *Crhr1* PFC transcripts in control ($n = 4$) and CUS ($n = 5$) mice as determined by RNA-seq (Likelihood ratio test with multiple comparisons; *Fos* FDR = 4.20E−08; *Fosl2* FDR = 1.65E−05; *Nr3c1* FDR = 0.04; *Nr3c2* FDR = 0.01; *Crhr1* FDR = 0.06). **l** Snapshot of genome browser depicting nuclear RNA-seq, YY1, and H3K27ac ChIP-seq reads at the mouse *Yy1* locus. Overlay of YY1 and H3K27ac ChIP signal ~5 kb upstream of the *Yy1* TSS is highlighted in a red dashed line box. **m** Snapshot of genome browser depicting nuclear RNA-seq, YY1, and H3K27ac ChIP-seq reads mapped to the mouse *Nr3c1* locus. Overlay of YY1 and H3K27ac ChIP signal ~20 kb upstream of the *Nr3c1* TSS is highlighted in a red dashed line box. **n** Levels of YY1 enrichment at the *Yy1* promoter region decrease in frontal cortices of CUS mice relative to controls (Mann–Whitney *U*-test; $P = 0.028$; $n = 4$ per group). **o** Levels of YY1 enrichment at the *Nr3c1* promoter region decrease in frontal cortices of CUS mice relative to controls (Unpaired *t*-test with Welch's correction; $P = 0.029$; $n = 4$ per group). *$P < 0.05$; **$P < 0.01$; ***$P < 0.001$; ****$P < 0.0001$. Error bars represent s.e.m. and statistical tests were two-sided unless stated otherwise. Source data are provided as a Source data file.

---

regulatory activity in CUS L2/3_Enpp2 neurons (Fig. 4c) prompted us to measure *Fos* and *Fosl2* expression in aCUS-exposed Yy1-exKO animals. We found that *Fos* and *Fosl2* are both downregulated in the PFCs of aCUS-exposed Yy1-exKO males compared to stressed Yy1-exGFP controls (Fig. 5j). In agreement with this finding, PFC tissues isolated from CUS males also showed significant decreased expression of these genes relative to non-stressed controls, providing additional evidence that loss of YY1 in PFC excitatory neurons induces a pattern of CUS-associated gene transcription (Fig. 5k).

Given that downregulated DEGs in L2/3_Enpp2 neurons isolated from CUS mice showed an enrichment of genes encoding synaptic proteins (Fig. 4a, b), we asked if aCUS exposure in YY1-ablated PFC excitatory neurons also modified the expression of synaptic genes. We assayed the expression of the synapse-related genes, *Adam23*, *Grm3*, and *Nrxn1*— which encode membrane-bound cell adhesion molecules and a glutamate metabotropic receptor—and found that their expression was significantly decreased in aCUS Yy1-exKO PFC tissues compared to controls (Supplementary Fig 7a). Given that these genes are not part of the YY1 regulatory network, these data indicate that loss of YY1 can indirectly impact the expression of synapse-related genes following stress exposure. Collectively with the aforementioned findings, these data demonstrate that an abbreviated 3-day CUS exposure in animals harboring selective loss of YY1 in PFC excitatory neurons provokes behavioral and transcriptional responses associated with 12 days of CUS exposure.

**YY1 directly regulates gene expression in the frontal cortex in response to chronic stress.** We next sought to assess whether the dysregulated expression of genes shared between Yy1-exKO aCUS and CUS animals could be directly caused, in part, by altered YY1 expression. We surveyed the genomic landscape at *Yy1* and *Nr3c1*—both of which were identified as members of the YY1 GRN/regulon in CUS L2/3 neurons (Supplementary Data 2)—for enrichment of YY1 occupancy using publicly available YY1 chromatin immunoprecipitation-sequencing (ChIP-seq) data obtained from E15 mouse cortical cells[55]. We reasoned that YY1 mediated the transcription of these genes through binding at YY1 motifs. We also examined H3K27ac, a chromatin mark associated with enhancers and active promoters which has been reported to co-localize with or near YY1 binding, in adult cortical excitatory neurons[56]. YY1 has been shown to autoregulate its own

expression by binding to a YY1 motif sequence in the first intron of the *Yy1* gene[57]. In agreement with this finding, we found an enrichment of YY1 binding, called as peaks by MACS2, aligned in the first intron near H3K27ac signal (Fig. 5l). YY1 reportedly regulates the expression of *Nr3c1* in the hypothalamus[47] and we reasoned that YY1 also mediated its transcription in cortical cells under stress conditions. Here, too, we observed enriched YY1 binding ~4 kb upstream of the *Nr3c1* transcription start site (TSS) that aligned with H3K27ac signal (Fig. 5m).

To determine if YY1 directly regulates *Yy1* and *Nr3c1* expression, we performed chromatin immunoprecipitation for YY1 in frontal cortical tissues dissected from the brains of control and CUS-subjected mice. This was followed by quantitative PCR (ChIP-qPCR) using primers designed against the genomic sequences near the *Yy1* and *Nr3c1* promoters with enriched YY1 binding (Fig. 5l, m). Notably, we observed a ~50% reduction of YY1 binding at the *Yy1* promoter in frontal cortices of CUS mice relative to controls, indicating that YY1 auto-regulates its gene expression in response to chronic stress ($P = 0.028$; Fig. 5n). YY1 binding was also significantly decreased at the *Nr3c1* promoter in CUS mice, consistent with the reduction in *Nr3c1* expression we observed in the mPFC ($P = 0.029$; Fig. 5k, o). Together these findings demonstrate a stress-responsive function for YY1 and a direct interaction between YY1 and its target genes that likely regulates their expression.

**CUS drives depressive- and anxiety-like behaviors in adult female mice that are mediated by YY1 activity in PFC excitatory neurons.** Given the increased prevalence of stress-related mood disorders in women, an examination of neuronal YY1 function and stress adaptation in female subjects is especially important. Accordingly, we first characterized the behavioral effects of CUS in adult female mice (9–10 weeks old) to confirm that the CUS paradigm drove the same depressive- and anxiety-associated phenotypes as was observed in male mice. Adult female mice (9–10 weeks) subjected to twelve consecutive days of CUS lost a significant amount of weight compared to age-matched controls (Fig. 6b), although body weights between the two groups were evenly distributed prior to CUS (Fig. 6a). Female CUS mice exhibited decreased food consumption (Fig. 6c), sucrose preference (Fig. 6d), and liquid consumption (Fig. 6e) relative to non-stressed controls. CUS female mice also spent significantly less time exploring the center of the open field arena

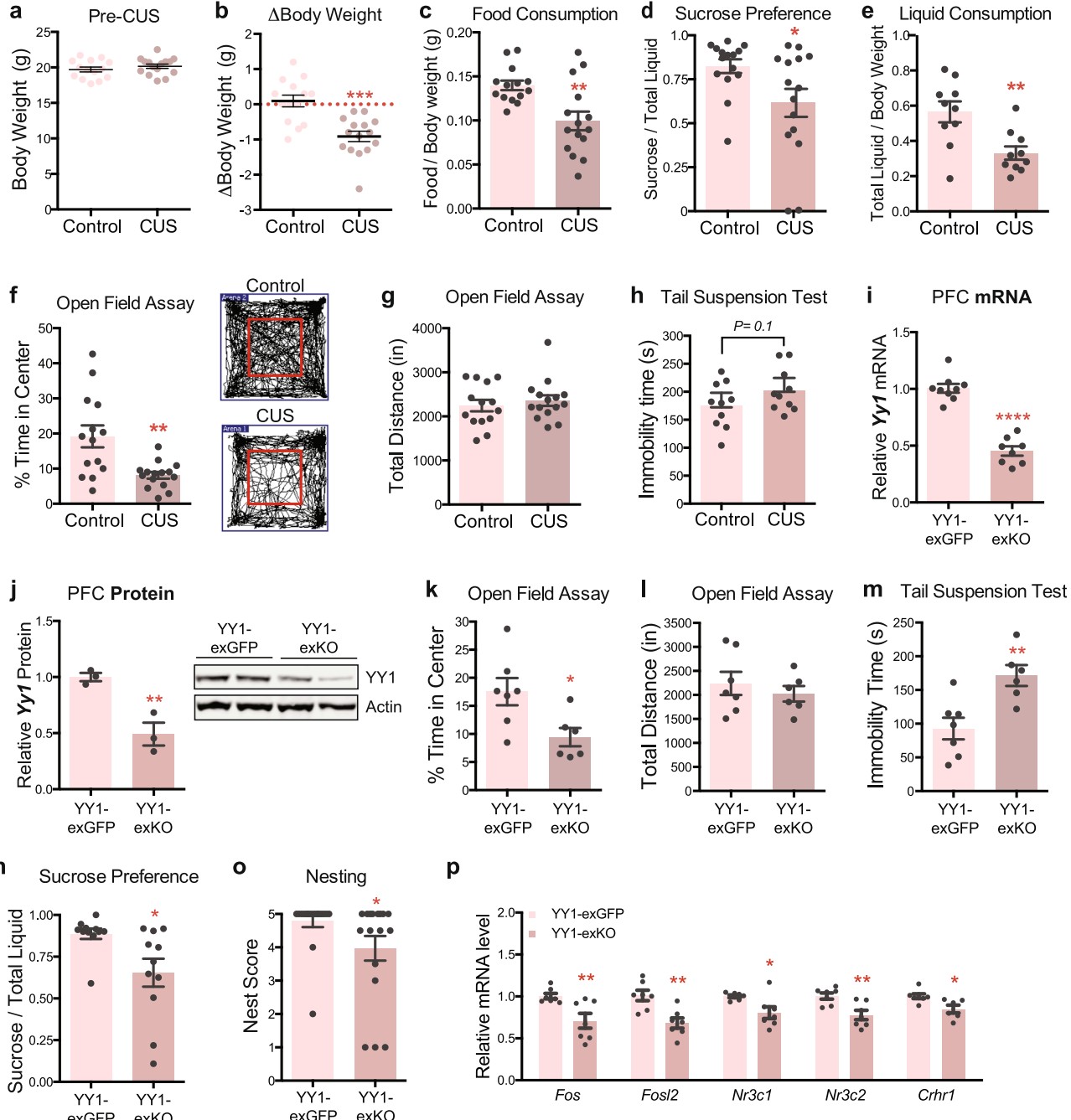

(Fig. 6f) although they traveled a comparable distance to unstressed controls (Fig. 6g). Additionally, CUS produced a modest effect on behavioral despair as measured by the tail suspension test; CUS females, like CUS males, showed a non-significant trend toward increased immobility compared to non-stressed controls (P = 0.1; Fig. 6h).

Having ascertained that CUS drives similar depressive- and anxiety-like phenotypes in male and female mice, we next assessed whether YY1 similarly functions in female PFC cortical neurons to control stress responses. We employed the same genetic strategy that we used in males to selectively inactivate YY1 from excitatory neurons in the PFCs of female mice (Supplementary Fig 6a). PFC tissues obtained from adult female *Yy1*<sup>fl/fl</sup> mice infected with AAV expressing *CamKII-eGFP-Cre* (YY1-exKO) showed a ~50% reduction in *Yy1* transcript (Fig. 6i) and protein levels (Fig. 6j) compared to *CamKII-eGFP* infected

females (YY1-exGFP). We had also observed a ~50% reduction of YY1 in AAV-infected male PFC tissues; thus YY1 levels do not significantly differ between males and females in this brain region.

We next assessed behavior in female Yy1-exGFP and Yy1-exKO mice. As we had observed in males, loss of YY1 in PFC excitatory neurons alone did not impair behavior in female animals. We observed comparable sucrose preference (Supplementary Fig 6f), nesting (Supplementary Fig 6g), exploratory behavior and locomotion in the open field test (Supplementary Fig 6h, i) between both YY1-exGFP and YY1-exKO cohorts. Remarkably, however, we found that selective deletion of YY1 in PFC excitatory neurons enhanced the stress sensitivity of adult females. YY1-exKO female mice, like their male counterparts, spent significantly less time in the center of the open field arena following exposure to aCUS than Yy1-exGFP controls (Fig. 6k)

**Fig. 6 CUS induces a depressive- and anxiety-like state in adult female mice that is driven by YY1 activity in PFC excitatory neurons. a** Pre-CUS body weights of control ($n = 14$) and CUS ($n = 15$) adult female mice. **b** CUS drives weight loss in adult female mice (Unpaired $t$-test; $P < 0.0001$; $n = 14$ controls, $n = 15$ CUS). **c** Food consumption of control and CUS females normalized to body weight. (Unpaired $t$-test with Welch's correction; $P = 0.003$; $n = 14$ controls, $n = 15$ CUS). **d** CUS decreases sucrose preference in adult female mice compared to unstressed controls (Mann–Whitney $U$-test; $P = 0.02$; $n = 15$ per group). **e** Liquid consumption of control and CUS females normalized to body weight (Unpaired $t$-test; $P = 0.004$; $n = 10$ per group). **f** CUS decreases exploratory behavior of adult female mice in the open field test (Unpaired $t$-test; $P = 0.004$; $n = 14$ controls, $n = 15$ CUS). **g** Locomotor activity of female mice is unaffected by CUS (Unpaired $t$-test; $P = 0.5$; $n = 14$ controls, $n = 15$ CUS). **h** Immobility times of control and CUS females subjected to the tail suspension test (Unpaired $t$-test; $P = 0.01$; $n = 10$ per group). **i** Quantification of *Yy1* mRNA levels in YY1-exKO mice ($n = 8$) relative to YY1-exGFP females ($n = 9$) (Unpaired $t$-test with Welch's correction; ****$P < 0.0001$). **j** Representative western blot of YY1 in medial PFC tissue lysates from YY1-exGFP and YY1-exKO females. Quantification of YY1 protein expression (normalized to β-actin) is shown on the right (Unpaired $t$-test; $P = 0.009$; $n = 3$ per group). **k** Decreased exploratory behavior in the open field arena exhibited by aCUS-exposed mice harboring selective loss of YY1 in PFC excitatory neurons (Unpaired $t$-test with Welch's correction; $P = 0.02$; $n = 7$ YY1-exGFP, $n = 6$ YY1-exKO). **l** aCUS does not alter locomotor activity of Yy1-exKO mice relative to Yy1-exGFP controls (Unpaired $t$-test; $P = 0.5$; $n = 7$ YY1-exGFP, $n = 6$ YY1-exKO). **m** Loss of YY1 in PFC excitatory neurons increases helplessness behavior in aCUS-exposed females in the tail suspension test (Unpaired $t$-test; $P = 0.005$; $n = 7$ YY1-exGFP, $n = 6$ YY1-exKO). **n** Loss of YY1 in PFC excitatory neurons decreases sucrose preference in aCUS-exposed females (Unpaired $t$-test; $P = 0.01$; $n = 11$ YY1-exGFP, $n = 11$ YY1-exKO). **o** Loss of YY1 in PFC excitatory neurons decreases nest scores in aCUS-exposed females (Unpaired $t$-test; $P = 0.01$; $n = 18$ YY1-exGFP, $n = 17$ YY1-exKO). **p** Quantitative RT-PCR measurements of *Fos, Fosl2, Nr3c1, Nr3c2*, and *Crhr1* mRNA levels in aCUS YY1-exKO females relative aCUS YY1-exGFP controls (Unpaired $t$-test; $P = 0.005$; $n = 7$ per group). *$P < 0.05$; **$P < 0.01$; ****$P < 0.0001$. Error bars represent s.e.m. and statistical tests were two-sided unless stated otherwise. Source data are provided as a Source data file.

without displaying altered physical activity (Fig. 6l) and exhibited increased behavioral despair in the tail suspension test (Fig. 6m). Notably, aCUS also drove a significant decrease in sucrose preference (Fig. 6n) and nest building scores (Fig. 6o), behaviors that had not been altered by aCUS in Yy1-exKO males (Fig. 5h, i).

Next, we asked if aCUS exposure in YY1-exKO female mice induced a similar transcriptional pattern of stress in the PFC as it did in males. We first measured the expression of the stress-related genes, *Nr3c1, Nr3c2*, and *Crhr1*, using RT-PCR on PFC tissues microdissected from aCUS-exposed YY1-exKO and YY1-exGFP females. We found significant reduction in the expression of *Nr3c1, Nr3c2*, and *Crhr1* in Yy1-exKO female PFC samples relative to controls (Fig. 6p). Moreover, expression of *Fos* and *Fosl2* were also significantly reduced in female Yy1-exKO PFC tissues (Fig. 6p), as was observed in males. We also compared levels of *Yy1* expression between male and female mice. Intriguingly, we found that *Yy1* was significantly more highly expressed in PFC tissues taken from female Yy1-exGFP animals exposed to aCUS than males, suggesting a pronounced role for YY1 in stress regulation in females (Supplementary Fig 7b).

Taken together, we conclude that twelve days of CUS induces stress-associated behaviors in both females and males and that these behaviors are, in part, mediated by YY1 function in PFC excitatory neurons.

## Discussion

In this study, we show that twelve days of chronic unpredictable stress induces a depressive- and anxiogenic-like state in male and female mice. By performing a battery of behavioral tests and genome-wide sequencing of nuclear RNA transcripts with cellular precision, we report a role for YY1 in mediating CUS-induced phenotypes in neocortical excitatory neurons. We also found altered patterns of chromatin interaction at the *Syt1* locus in CUS-subjected mice that were associated with neuronal inactivation, implicating dynamic changes in chromatin folding in response to stress exposure. We report that neocortical excitatory neurons—in particular layer 2/3 pyramidal neurons—exhibited the most transcriptional dysregulation by CUS, underscoring an enhanced sensitivity to chronic stress as well as a function for L2/3 neurons in modulating stress effects on behavior[58]. Furthermore, we found that CUS drove a significant downregulation of synaptic genes in L2/3 cortical neurons, providing mechanistic support to previously described findings of decreased spine

density and volume that have been observed in both post-mortem PFC tissues of depressed humans[59] and stressed rodents[14,16,17,60].

YY1 is a ubiquitously expressed zinc finger protein that contributes to structural enhancer-promoter interactions[43,61], a common feature of mammalian gene control. YY1 plays a critical role in cortical development and haploinsufficiency of YY1 causes "YY1 syndrome", a neurodevelopmental disorder characterized by intellectual disability, seizures, and behavioral impairment[62]. Despite its well-characterized role in early development, relatively little is known about YY1's function in the adult brain. We found that *Yy1* expression is reduced in cortical excitatory neurons by chronic stress in single-nucleus RNA-sequencing experiments, but not in bulk RNA-seq analyses (logFC = −0.8; FDR = 0.5), underscoring the cell type-specific effect of this dysregulation in this heterogeneous brain region. Specific loss of YY1 in PFC excitatory neurons impairs stress coping ability in both male and female mice, as measured by transcriptional deregulation of stress-associated genes in the prefrontal cortex and stress-associated behavioral phenotypes.

Our findings complement a previous genomics study of whole blood samples from MDD patients that identified YY1 as an upstream regulator of an MDD-associated transcriptional program. That study also found that *Yy1* expression is negatively correlated with MDD status[45]. Together, this work indicates that decreased YY1 function is not only associated with depressive-like behaviors in PFC excitatory neurons of chronically stressed mice but may also influence the MDD disease course in humans. Furthermore, most studies to date have virally manipulated gene expression in vivo by infecting brain regions without cell type selectivity to functionally validate the role of a target gene on behavior, thereby obscuring potential cell type-specific effects and contributions to the observed phenotype. In our study, we used *CamKII*-promoter driven expression of Cre recombinase to selectively delete *Yy1* from PFC excitatory neurons and demonstrate that YY1 function in this cell type, without manipulation of inhibitory neurons and glia, enhances stress sensitivity in vivo.

We show that inactivation of YY1 in PFC excitatory neurons alone did not impair behavior in male and female mice but rendered them more sensitive to stress. Although YY1 is constitutively expressed into adulthood, a recent study has shown that YY1 function is most critical during early cortical development and that neuronal dependence on YY1 decreases with age[55]. Our study demonstrates that loss of YY1 in cortical excitatory neurons alone does not drive functional impairment of the PFC

but impairs neuronal adaptation to environmental stimuli. Specifically, loss of YY1 itself does not deregulate behavior in male and female mice under homecage conditions but rather incapacitates their ability to appropriately respond to stress—these findings suggest that YY1 dynamically functions to maintain neuronal homeostasis.

Compiling data from our bulk and single nuclei RNA-seq and 5C experiments also provides insight into the neuronal activity patterns induced by stress exposure. Computational reconstruction of gene regulatory networks from L2/3 sNuc Drop-seq data uncovered an upregulation of regulons governed by activity-dependent transcription factors, including Fos, Fosb, and Egr1 (Fig. 4c), indicating that the genes these transcription factors regulate are elevated in L2/3 neurons after 12 days of stress. In contrast, bulk nuclei RNA-seq revealed that the transcription of many activity-regulated genes, including *Fos* and *Fosb*, are downregulated in the PFCs of CUS mice (Fig. 2g). Likewise, GO analysis of the upregulated DEGs from bulk nuclear sequencing of CUS mice showed an enrichment of receptor-activity and cell-signaling terms (Fig. 2e), consistent with prior neuronal activation, while higher-order genome organization in these tissues were also found to have reorganized in a pattern associated with neuronal silencing (Fig. 2h). Together, these data portray a biphasic and time-dependent neuroadaptive response to stress, in which neuronal activation of the PFC may have occurred early during the CUS paradigm, leading to a remodeling of chromatin configuration and transcription in PFC excitatory neurons to decrease activity following sustained CUS exposure. The juxtaposition of these findings bridge together previously published work demonstrating that acute stress activates glutamatergic neurotransmission in the PFC[63,64] to others that have shown opposing effects in the PFC after long-term prolonged stress, namely dampened excitatory transmission and atrophied dendritic architecture[14–18].

Our study also uncovered sex-discordant responses to aCUS in the sucrose and nesting assays. aCUS-subjected Yy1-exKO females showed decreased sucrose preference and nesting while the behavioral responses of YY1-exKO males in these same behavioral tests were virtually indistinguishable from Yy1-exGFP controls. We also found that *Yy1* expression in the PFC is increased in female animals subjected to aCUS compared to their male counterparts (Supplementary Fig 7b). These data intimate that YY1 inactivation in stress-exposed neurons affects female PFC function more broadly than in males to disrupt additional domains of behavior and point to a sex-biased function for YY1 in stress regulation that may, in part, underlie the increased incidence of stress-related mood and anxiety disorders in women. Future studies directly comparing male and female responses to aCUS, which were not performed in this study, are needed to address whether YY1-exKO females show enhanced stress-induced behavioral impairment than their male counterparts. Additionally, while our study validated YY1's functional role in neuronal stress adaptation, it did not characterize the effects of YY1 overexpression on stress susceptibility. However, a recent study characterizing transcriptional regulation of stress resilience identified an enrichment of YY1 motifs in the promoters of stress resilience-associated genes[65]. That finding supports this study, and, collectively, provide rationale for future work investigating YY1's ability to rescue stress-induced behavioral phenotypes.

Intriguingly, our study found a sex-specific response in *Crhr1* gene expression between aCUS-exposed Yy1-exKO male and female mice. While aCUS increased *Crhr1* mRNA levels in PFC tissues of Yy1-exKO males relative to Yy1-exGFP controls, it led to decreased *Crhr1* expression in Yy1-exKO females. This dimorphic effect on *Crhr1* may be due to a sex-specific function of the CRH system in the PFC. Previous work demonstrates the presence of higher basal CRH levels in female PFCs than in males as well as sex-specific behavioral outcomes to CRH activation of neural circuits in the PFC[66]. We also observed significantly higher levels of *Crhr1* in the medial PFC of female mice compared to age-matched male littermates (Supplementary Fig 7c). This disparity in the expression and function of the CRH system in the PFC likely underlies the different effects that aCUS exerts on *Crhr1* expression in Yy1 exKO males and females.

Our study highlights the critical role of epigenetic factors in mediating cellular responses to environmental stimuli, which is an especially vital process in post-mitotic neurons. We show that YY1 activity is essential for neuronal adaptation to stress in both males and females, underscoring its generalizability as a target for therapeutic treatment. Our data also provide evidence for other factors involved in organizing chromatin structure, such as CTCF, in the stress-induced nuclear reprogramming of cortical neurons; these factors may function in concert with YY1 in re-shaping the transcriptional landscape in response to stress. Taken together, our findings show that chronic stress exerts a significant impact on PFC excitatory neurons to decrease their activity, and highlights the role of the nucleus as the dynamic center of coordinated cellular activity driving the adaptive transcriptional responses to chronic stress that ultimately modify behavior. Our study also demonstrates ethological validation of the CUS rodent model and its utility in uncovering clinically-relevant mechanistic insights into the behavioral consequences of stress.

## Methods

**Animals.** Mice used for CUS experiments were C57Bl/6J mice obtained from Jackson Laboratories at 8 weeks of age and given 1.5–2 weeks of acclimation before CUS. *Yy1^fl/fl* mice[49] were obtained from M. Atchison and maintained on a *C57BL/6J* background. All mice were maintained on a 12 h light-dark cycle with lights on at 7:00 a.m. Food and water were provided ad libitum except during one overnight 13 hr period when food was removed during the chronic unpredictable stress paradigm. All experiments conformed to the Institutional Animal Care and Use Committee (IACUC) guidelines at the University of Pennsylvania.

**Chronic unpredictable stress.** CUS was performed in 9–10 weeks old mice as described in Supplementary Table 1 and as previously described[27]. Three different stressors were performed each day for variable lengths of time over a 12-day period in dedicated procedure rooms. The first CUS stressor began at 8 a.m., the second at 1 p.m., and the last stress began at 6 p.m. and continued until 7 a.m. the following morning. Mice were group housed (4–5 mice per cage) and controls were housed in a separate room from CUS animals and gently handled daily during the duration of CUS. Control and CUS mice that were used for behavioral experiments were singly housed beginning at 6 p.m. on the 12th final day of the CUS paradigm for sucrose preference, food consumption, and nesting assays.

**Abbreviated CUS.** aCUS was performed as described in Supplementary Table 2 over the course of 3 days, with three different stressors performed each day as in CUS. Mice that were used for behavioral experiments were singly housed beginning at 6 p.m. on the third final day of the CUS paradigm for sucrose preference testing and nesting assays.

**Animal behavior.** All behavioral testing was counterbalanced across experimental groups and mice were randomly assigned to each group as previously described[67]. Behavioral analysis for open field, elevated zero maze, and tail suspension tests was performed automatically in real-time by video-tracking software (SmartScan 3.0). With the exception of the tail suspension test, every other behavioral test took place in a dedicated room with low, indirect lighting apart from where CUS was performed. All behavioral testing equipment was wiped clean between animals to remove odor cues, first with Versa-clean (Fisherbrand) diluted 1:80 in deionized water and then with deionized water alone. Experimenters changed gloves between handling control and CUS animals and between handling male and female mice. Experimenters were never present in the room during behavioral testing. With the exception of the sucrose preference test, all behavioral testing started at 7:00 a.m., when the light cycle turned on, and mice were given 1 h to habituate to a new room prior to testing.

*Sucrose preference*, *food consumption*, and *nesting* assays began at 6 p.m. on the final day of CUS. Control and CUS mice were singly housed in new cages with one intact nestlet, one pre-weighed food pellet, and two pre-weighed bottles, one filled with the animal's drinking water and the other containing sucrose dissolved in the

animal's drinking water to a concentration of 2.5%. Details for each assay are included below:

*Food consumption*. Food pellets were collected from each cage at 7:00 am the next morning, 13 h after mice were singly housed. Food consumption was measured as the change in weight of the food pellet.

*Nesting*. Nest construction was evaluated using the metric as described previously[67]. 18 h after mice were singly housed with a cotton square nestlet (Ancare) and no other bedding material. Nests were assessed for amount of nestlet material shredded, height, and shape and scored using the following metric: (1) nestlet not noticeably touched; (2) nestlet partially torn; (3) nestlet mostly shredded but with no identifiable nest site; (4) an identifiable but flat nest; and (5) a perfect nest with walls. A score of 4.5 was given to nests that had walls covering less than 50% of the nest circumference.

*Sucrose preference test*. Animals were introduced to two 50 ml tubes (Falcon) filled with their drinking water and plugged with a rubber stopper holding a drinking tube (Ancare) 24 h prior to the sucrose preference test. At 6 pm on the final day of CUS, control and CUS mice were singly housed in new cages containing pre-weighed water and 2.5% sucrose bottles. The positions of the sucrose and water bottles were switched 12 h into testing, and collected for weighing 24 h after testing was initiated. Sucrose preference was calculated as change in weight of the sucrose bottle (sucrose consumption) divided by change in weights of both sucrose and water bottles (total liquid consumption). Mice were re-grouped following sucrose preference testing and remained group-housed for the duration of the behavioral testing.

*Coat state*. Coat scores were assigned for each mouse and assessed by assigning a score of 0 (clean, sleek coat), 0.5 (fur moderately or partially deteriorated or dirty) or 1 (fur mostly or entirely deteriorated, dirty, and/or raised) to eight different body regions of the mouse. Scores were summed to obtain severity of coat state scores for each mouse.

*Open field assay*. Mice were given 15 min to explore a $15'' \times 15''$ box placed directly underneath a ceiling-mounted camera. Center and periphery of the arenas were defined in video tracking software and percent time spent in the center and total distance traveled was recorded and used for analysis.

*Elevated zero maze*. Mice were placed inside a designated closed arm of an elevated zero maze (San Diego Instruments) directly facing the open arm and allowed to explore the maze for 5 min. Percent time spent inside the closed and open arms of the maze and total distance traveled were recorded by video tracking software. The video-tracking results were later manually validated by two blinded experimenters who were trained to analyze percent time for each mouse in the open arms using recorded videos.

*Tail suspension test*. Mice were suspended inside a pre-calibrated BIOSEB tail suspension apparatus (BIO-TST5) by attaching tape placed over the ends of their tails to a sensor on the ceiling of each chamber. $1.5''$ plastic cylinders were threaded onto their tails prior to testing to prevent tail climbing behavior. Immobility time was recorded for each mouse over a 6-min period using the automated BIOSEB software. The software-tracking results were later validated by one blinded, trained experimenter who manually recorded immobility times on video recordings of a subset of randomly chosen mice.

**Nuclei isolation and purification**. Mouse neocortical tissues were collected from a cohort of mice 24 h after the final CUS stressor for sNucDrop-seq. Cortical tissues were rapidly resected on ice, flash-frozen in liquid nitrogen, and stored at $-80\,°C$ before nuclear isolation. Cortical nuclei were isolated as previously described[30]. Briefly, tissues were dounce homogenized on ice in 12 mL of sucrose homogenization buffer containing 0.32 M sucrose (Sigma-Aldrich, RNase & DNase free, ultra pure grade), 5 mM CaCl$_2$ (Sigma-Aldrich), 3 mM MgAc2 (Sigma-Aldrich), 10 mM Tris-HCl pH 8.0 (Invitrogen), 0.1% Triton X-100 (Sigma-Aldrich), 0.1 mM EDTA (Invitrogen), and protease inhibitor cocktail (Roche), gently layered on top of 14 mL of sucrose cushion buffer (1.8 M sucrose, 10 mM Tris-HCl pH 8.0, 3 mM MgAc2, protease inhibitor cocktail) in a $1'' \times 3.5''$ sterile centrifuge tube (Beckman), and isolated by ultracentrifugation at 25,000 rpm at 4 °C for 2 h using a Beckman Coulter SW28 swinging bucket rotor. For bulk nuclear RNA-seq experiments, nuclei were resuspended in 1 mL of ice-cold PBS containing EDTA-free protease inhibitor cocktail tablet (Roche), and RNasin ribonuclease inhibitor (Promega) and transferred to a 1.5 mL tube. For sNuc Drop-seq nuclei were resuspended in 1 mL of ice-cold PBS containing 0.01% BSA (Sigma-Aldrich), EDTA-free protease inhibitor cocktail tablet (Roche), and RNasin ribonuclease inhibitor (Promega) and transferred to a 1.5 mL tube.

**Single-nucleus Drop-seq and data analysis**. sNucDrop-seq of cortical nuclei was performed as previously described[36]. Briefly, nuclei suspensions were run through an Aquapel-coated PDMS microfluidic device (uFluidix) with barcoded beads (ChemGenes) to co-encapsulate individual nuclei with a single bead. Barcoded beads were resuspended in lysis buffer (200 mM Tris-HCl pH 8.0, 20 mM EDTA, 6% Ficoll PM-400 (GE Healthcare/Fisher Scientific), 0.2% Sarkosyl (Sigma-Aldrich), and 50 mM DTT (Fermentas) at a concentration of 120 beads/**u**L. Droplet breakage with Perfluoro-1-octanol (Sigma-Aldrich), reverse transcription using Maxima H Minus Reverse Transcriptase (ThermoFisher) and exonuclease I treatment were subsequently performed. cDNA was amplified by PCR (KAPA HiFi hotstart Readymix, KAPA biosystems) using a pre-determined, optimized number of cycles and purified twice with 0.6X SPRISelect beads (Beckman Coulter). cDNA was then tagmented using the Nextera XT DNA sample preparation kit (Illumina, cat# FC-131-1096) and further amplified using 12 enrichment PCR cycles. Libraries were sequenced on an Illumina NextSeq 500 using the 75-cycle High Output v2 Kit (Illumina), each loaded at a concentration of 2.0 pM. In total, 8 individual mouse cortex samples (4 controls, 4 CUS) were analyzed by sNucDrop-seq with 2 independent batches (2 controls, 2 CUS samples per batch), which were sequenced twice.

*Read mapping, clustering, and marker gene identification*. Paired-end sequencing reads of sNucDrop-seq were processed as previously described[36]. In brief, after mapping the reads to the mouse genome (mm10, Gencode release vM13), both exonic and intronic reads mapped to the predicted strands of annotated genes were retrieved for the cell type classification. Uniquely mapped reads were grouped by cell barcode. To digitally count gene transcripts, a list of UMIs in each gene, within each nucleus, was assembled, and UMIs within ED = 1 were merged together. The total number of unique UMI sequences was counted, and this number was reported as the number of transcripts of that gene for a given nuclei. Raw digital expression matrices were generated for the 4 Nextseq 500 sequencing runs.

The raw digital expression matrices of libraries from control and CUS mice were combined and loaded into the R package Seurat[68] (version 3.1.1.9002). For normalization, UMI counts for all cells were scaled by library size (total UMI counts), multiplied by 10,000 and transformed to log space. Only genes found to be expressing in >10 cells were retained. Cell with a high percentage of UMIs mapping to mitochondrial genes ($> = 0.05$), fewer than 600 detected genes, or more than 5,000 detected genes were discarded. In addition, doublet analysis tool, Scrublet[69] (version 0.2), was used to remove cells with doublet score >0.2 for each sample. As a result, 31,806 nuclei from 8 samples (4 control and 4 CUS) were kept for downstream analysis. The top 2000 highly variable genes (HVGs) were identified using the function *FindVariableFeatures* with "vst" method. The expression levels of HVGs in the nuclei were scaled and centered along each gene and was subjected to PCA analysis. Assessing a number of different PCs for clustering revealed that the variation of PC number selection was relatively insensitive to the clustering results. The top 50 PCs were selected and used for 2-dimension reduction by Uniform Manifold Approximation and Projection (UMAP). Clusters were identified using the function *FindCluster* in Seurat with the resolution parameter set to 1. Cells were classified into 21–45 clusters with the resolution parameter from 0.3 to 2. Clustering resolution parameters were varied quantitatively based on the number of cells being clustered. After the clustering results with different resolutions were compared and evaluated, we chose a resolution value of 1. Using this approach we were able to assign 31,806 cells to 28 clusters. Marker genes were then identified using the function *FindAllMarkers* in Seurat. Cell type was annotated based on top ranked marker genes. Two cell clusters, which co-expressed multiple cell type specific markers, were empirically considered as doublets. In all, 546 nuclei (1.7% of input data) were removed from the downstream analysis and 31,206 cells were finally assigned into 26 cell clusters (Fig. 3a and b).

*Identification of cell-type-specific differentially expressed genes between Control and CUS nuclei*. In this study, we randomly selected 4 mice (2 control and 2 CUS) from each batch, in total 8 animals from 2 batches. After quick assessment of transcriptional signature of control and CUS mice by MDS plot, we found a profound batch effect in the differential expression gene test. To account for the batch effect in gene expression, edgeR[70] generalized linear model (GLM) was used to analyze the gene expression difference under different group (Control or CUS) and different batch. 26 cell clusters with more than 200 nuclei were subjected to differential expression test. For each cluster, raw gene counts were aggregated by gene and animal identity. To filter out low expressed genes and compensate for different nuclei size of clusters, we kept the genes whose aggregated UMI was more than 10% of nuclei number in a cluster. By doing this, we obtained comparable number of genes that were subjected into the differential expression test across clusters, varied from 11,336 genes to 14,856 genes. Then model.matrix function in edgeR was used to construct the design matrix with two factors, group and batch. Then likelihood ratio test by functions glmFit and glmLRT was performed to test the differential expression of genes between Control and CUS mice. The one-sided likelihood ratio test was performed to test the significance of differentially expressed genes between Control and CUS mice. P-value was then corrected by FDR. Genes with FDR < 0.2 were considered significantly expressed between Control and CUS nuclei (Fig. 3c).

*Gene ontology enrichment analysis*. GO enrichment analysis were performed as previously described[30]. To identify functional categories associated with defined

gene lists, the GO annotations were downloaded from the Ensembl database. An enrichment analysis was performed via a hypergeometric test. The *P* value was calculated using the following formula:

$$P = 1 - \sum_{i=0}^{m-1} \frac{\binom{M}{i}\binom{N-M}{n-i}}{\binom{N}{n}} \qquad (1)$$

where *N* is the total number of background genes, *n* is the total number of selected genes, *M* is the number of genes annotated to a certain GO term, and i is the number of selected genes annotated to a certain GO term. *P* value was corrected by function p.adjust with false discovery rate (FDR) correction in R. GO terms with FDR below 0.05 were considered enriched. All statistical calculations were performed in R.

*SCENIC analysis.* To assess the regulatory activity of transcript factors associated with CUS, we used SCENIC[42] (version 1.1.2.2) to perform gene regulatory network analysis. Regulatory modules are identified by inferring co-expression between TFs and genes containing TF binding motif in their promoters. We took out nuclei with relatively high number of detected genes (>1000 nGenes) for SCENIC analysis from Ex_L2/3_Enpp2 cluster. 1788 control nuclei and 1933 stressed nuclei from 8 mice were used for downstream analysis. Two gene-motif rankings, 10 kb around the TSS and 500 bp upstream, were loaded from RcisTarget databases (mm9). Gene detected in >1% of all the nuclei and listed in the gene-motif ranking databases were retained, resulting in 10,840 genes. Then GRNBoost, which was implemented in pySCENIC, was used to infer the co-expression modules and quantify the weight between TFs and target genes. Targets genes that did not show a positive correlation (>0.03) in each TF-module were filtered out. SCENIC found 5892 TF-modules. A cis-regulatory motif analysis on each of the TF-modules with Rcis-Target revealed 346 regulons (transcription factors). The top 1 percentile of the number of detected genes per cell was used to calculate the enrichment of each regulon in each cell. The regulatory activity was quantified by area under the recovery curve (AUC) value from the enrichment of each regulon.

AUC values of TFs were obtained and then subjected to Wilcoxon rank sum test to access significance of the difference of TF activity between CUS and control nuclei per batch. TFs with *P*-value less than 0.05 and showing consistent trend in 2 batches were considered differentially regulated. For Fig. 4c, we computed the mean AUC of all nuclei belonging to defined groups, then normalized CUS to Control group and log-transform the fold change. R package pheatmap was used to draw the heatmap. For Fig S3, box plot was used to show the AUC value of transcription factors at different batch and group.

**Bulk nuclear RNA-seq and data analysis.** Mice for bulk nuclear RNA-seq analysis were taken 24 h after the last behavioral test was conducted. Medial PFC tissues were isolated using a sterile 2 mm disposable biopsy punch (Integra Miltex) centered over the midline of 1 mm-thick coronal section cut using a brain matrix (Harvard Apparatus). Each sample was flash-frozen in liquid nitrogen and stored at −80 °C before nuclear isolation. Nuclei were pelleted at 5000 rpm for 10 min at 4 °C. The supernatant was subsequently removed, and total nuclear RNA was isolated by Trizol (Thermo Fisher) and treated with DNase I (Thermo Fisher). Samples were prepared for RNA-sequencing using the TruSeq Stranded Total RNA library prep kit with RiboZero depletion (Illumina). Multiplexed libraries were submitted for sequencing on the Illumina HiSeq 2500 platform at the University of Pennsylvania Next-Generation Sequencing Core facility. FASTQ files for each RNA sequencing library were mapped to the mouse mm10 genome by STAR using the parameters of '--outFilterMultimapNmax 1 --outFilterMismatchNmax 3'. The read number of each gene was counted by in-house Perl programs as previously described[30,71] to normalize raw counts and to compare between control and CUS samples. We used edgeR 3.22.5 in R version 3.5.2 to compare the gene expression profiles between control and CUS samples. We used the "glmFit" function in egdeR to fit a negative binomial generalized log-linear model to the counts for each gene. We used the "glmLRT" function in edgeR to conduct likelihood ratio tests in the linear model using the parameter of "coef = 2". The tests were adjusted for multiple comparisons. Genes with a false discovery rate (FDR) less than 0.05 were considered as differentially expressed genes as described in previous studies[30,71].

*GSEA.* A pre-ranked GSEA was performed with the use of GSEA v.4.0.3 using a gene set consisting of neuronal primary response genes identified by Tyssowski et al.[32]. Enrichment of these genes was analyzed in a list of transcripts identified by RNA-seq analysis that were ranked by differential expression with 1000 permutations to assess the statistical significance of the enrichment score.

**RNAScope in situ hybridization.** Mice were anesthetized via intraperitoneal (IP) injection of ketamine cocktail followed by perfusion with 4% paraformaldehyde (PFA). Brain tissues were then isolated for post-fixation in 4% PFA overnight at 4 °C, cryo-preservation in 15% sucrose, then 30% sucrose, overnight at 4 °C, and finally embedded in Tissue-Tek OCT compound (Sakyra Finetek) and stored in −80 °C. Coronal sections (7 μm thickness) were collected on glass slides (Fisher

Scientific) using a Cryo-Stat, dried at room temperature (RT) for 2 h, and stored at −20 °C prior to staining following RNAScope protocol.

RNAScope in situ hybridization [Advanced Cell Diagnostics (ACD)] was performed according to the manufacturer's instructions. OCT imbedded tissue sections (7 μm) on glass slides were immersed in a prechilled 4% PFA for 15 min. After rinsing with sterile PBS twice, gradient dehydration using 50, 70, and 100% ethanol was performed for 5 min per step. Then, dehydration was repeated in 100% anhydrous ethanol followed by drying at RT for 10 min. Afterward, sample slides were incubated with RNAScope hydrogen peroxide (ACD, REF:322381) for 10 min at RT and then rinsed three times with sterile PBS. Sample slides were next incubated with 200 ml of RNAScope 1× Target Retrieval Reagent (ACD, REF:322000) at 100 °C for ~5 min, followed by washing immediately with distilled H₂O for 15 s, 100% alcohol for 3 min, and air dry. Dried slides were then incubated with Protease III for 30 min at 40 °C, rinsed in sterile water with slight agitation for 2 min, incubated with five drops of individual probe mix to cover each section entirely. The target probes were Yy1 (Mm-Yy1, REF:575381, ACD) in channel C1 and Cdkl5 (Mm-Cdkl5-C2, REF:500851-C2, ACD) in channel C2. Sample slides were finally washed for 2 min three times with the wash buffer (REF:310091) at RT, incubated with five drops of RNAscope Multiplex AMP 1 (40 °C for 30 min), AMP 2 (40 °C for 30 min), and AMP3 (40 °C for 15 min). Channel C1 was illustrated with fluorescent dye (OpalTM 570 reagent pack, FP1488001KT, PerkinElmer) as red and channel C1 was illustrated with fluorescent dye (OpalTM 690 reagent pack, 1497001KT, PerkinElmer) as gray. All samples were counterstained with DAPI prior to image acquisition.

*Image analysis.* Confocal images were captured with Leica TCS SP8 confocal microscope at × 63 magnification. Collected images were analyzed using ImageJ (NIH). Individual spots of RNAs were counted, blind of sample IDs, for the expression levels of Yy1 and Cdkl5 at c1 or c2 channels, respectively, on the same cell/slide. Three images were randomly selected from L2/L3 and L6 brain sections of each animal. 10 cells per image for a total of 3 mice in each control or CUS condition were collected for data analysis.

**Primary neuronal culture.** Murine primary cortical neurons were purchased from the Neurons-R-Us Core at the University of Pennsylvania Perelman School of Medicine. Approximately 300,000 cells were plated into each well of a 12-well plate for RT-PCR and 600,000 cells into each well of a 6-well plate for western blotting. Cell cultures were maintained in neurobasal medium supplemented with B27, 1× Glutamax (Gibco, Thermo Fisher Scientific), 0.5% D-glucose, and 100 μg/ml Primocin (InvivoGen), with 50% media changes every other day for a 7-day period. The 1.5-week corticosterone (CORT) treatment began on DIV7. Half of the media in a subset of wells was exchanged with fresh media mixed with 2 μM corticosterone for a total concentration of 1 μM CORT while the rest of the wells were treated with fresh media mixed with an equivalent amount of vehicle (DMSO). Media was exchanged on all cells simultaneously at every time-point so that CORT-treated cells received fresh CORT at every subsequent time-point. Cells for all time-points were harvested simultaneously for RT-PCR and western blotting. Nuclei were isolated from cultured cells for RT-PCR and whole-cell lysates were used for protein analyses. For neuronal activity modulation, cortical neurons from E18 WT C57/BL6 mouse embryos were dissociated and plated at a density of 200,000 cells/mL. At DIV15 neurons were treated for 24 h with either 1 μM Tetrodotoxin (TTX) or 10 uM Bicuculline (Bic) via addition to the cell culture media and harvested for 5C.

**Viral reagents and surgeries.** EGFP or eGFP fused to Cre recombinase was expressed in $Yy1^{fl/fl}$ mice using AAV9.CamKII.eGFP or AAV9.CamKII.eGFP-Cre obtained from the Penn Vector Core and diluted to $1.0 \times 10^{13}$ per mL in sterile PBS before use.

Stereotaxis-assisted intra-PFC injections were performed in $Yy1^{fl/fl}$ mice at 10–14 weeks of age. Mice were deeply anesthetized with isoflurane by inhalation, placed in a small-animal stereotaxic device with bilateral arms (Kopf Instruments), and treated pre-operatively with lidocaine at the injection site and meloxicam (5 mg/kg) subcutaneously. After the skull surface was exposed and leveled, two custom-made 16 mm long 33-gauge needles (Hamilton) were simultaneously lowered at pre-drilled holes at the following coordinates from Bregma: +1.8 A/P; ±0.8 M/L, −2.75 D/V, at 15° angle from the midline. Viral suspension (300 nL) was infused over a 3-min period (100 nL/min) and needles were kept in place for 5-min after injection to prevent flowback. Mice were treated with a subcutaneous injection of meloxicam (5 mg/kg) 24 h following surgery and monitored 3 times a week over the 3-week recovery period. In vivo transduction was confirmed on dissected PFC tissue by RT-PCR and GFP expression under a fluorescent microscope.

**Quantitative RT-PCR.** Medial prefrontal cortical tissue was rapidly microdissected using the method described above. RNA was isolated by Trizol reagent (Thermo Fisher), treated with DNase I (Thermo Fisher), and purified using the RNeasy MinElute Clean-up kit (Qiagen cat. #74204). Thousand nanogram of RNA was converted into cDNA using the High-Capacity cDNA Reverse Transcription Kit (Applied Biosystems, 4368814) and real-time-PCR was performed using Taqman

Gene Expression Assay probes purchased from Applied Biosystems and TaqMan Universal PCR Master Mix (Applied Biosystems, cat. #4304437). The following Taqman assay primer/probe sets were used for this study: Hprt (Mm03024075_m1); Yy1 (Mm00456392_m1), Nr3c1 (Mm00433832_m1), Nr3c2 (Mm01241596_m1), Crhr1 (Mm00432670_m1), Fos (Mm00487425_m1), Fosl2 (Mm00484442_m1), Adam23 (Mm00478606_m1), Grm3 (Mm00725298_m1), Homer1 (Mm00516275_m1).

Results were quantified on an ABI 7900 system. All RNA expression levels were normalized to *Hprt* using the ΔΔCT method.

**Western blotting.** Medial prefrontal cortical tissue was isolated using the methods described above and homogenized in RIPA buffer (1% Triton X-100, 1% sodium deoxycholate, 150 mM NaCl, 0.1% SDS, pH 8.0) with protease inhibitors (Roche, cOmplete, EDTA-free protease inhibitor cocktail tablets). Homogenized lysates were incubated on ice for 15 min, and then centrifuged at $21,000 \times g$ for 15 min at 4 °C. The supernatant fraction was sonicated using a Biorupter three times at max frequency for 15 s, each followed by a 60 s cooldown period. The lysate was then centrifuged at $21,000 \times g$ for 15 min at 4 °C, and the supernatant was quantified using the BCA Protein Assay Kit (Pierce, Rockford, IL) for Western blotting. Equal amounts of protein lysate (30 μg) were run and separated on a 10% SDS–PAGE gel (Invitrogen) and transferred to a polyvinylidene fluoride membrane (Millipore). The resulting membrane was blocked with a 1:1 solution of Odyssey blocking buffer (LI-COR; 927-40100) for 1 h at room temperature and incubated with the following antibodies overnight: mouse anti-β-Actin (Abcam; ab8226; diluted 1:10,000); mouse anti-YY1 (Santa Cruz; H-10, sc-7341; diluted 1:1,000); rabbit anti-YY1 (Cell Signaling; D5D9Z Rabbit mAb #46395; 1:1,000). Secondary antibodies (LI-COR) used were anti-mouse IgG IRDye680LT and incubated for 1 h at room temperature at dilutions of 1:10,000. Standard protocols were used for the Odyssey Infrared Imaging System (LI-COR) for protein visualization and quantification. Uncropped blots are provided in Source data.

**5C.** PFC tissues were rapidly microdissected from frontal cortical slices cut in a pre-chilled brain matrix (Harvard Apparatus) and immediately snap-frozen in liquid nitrogen. Frozen tissue was processed for in situ 3C as previously described[72]. Briefly, frozen PFC tissue was placed in Covaris TT1 tissue tubes and pulverized using the CP01 Cryoprep Manual Pulverizer, following the manufacturer's instructions. Tissue was transferred from tissue tubes to 15 mL conical tubes using 10 mL cold PBS, to which 1 mL of fixation solution (50 mM Hepes-KOH, 100 mM NaCl, 1 mM EDTA, 0.5 mM EGTA, 11% formaldehyde) was added following by 10 min of rotation at room temperature. Fixation was terminated via the addition of 0.58 mL 2.5 M glycine and 5 min of room temp incubation, followed by 2× PBS washes. Fixed cells from two animals were pooled before proceeding with in situ 3C as previously described[61,73].

5C primers were designed according to the double-alternating design scheme[74] using the My5C primer design software (http://my5c.umassmed.edu/my5Cprimers/5C.php)13 with universal "Emulsion" primer tails. 5C reactions were carried out as previously described[73]. Libraries were evenly pooled and sequenced on the Illumina NextSeq 500 using 37 bp paired-end reads. Reads were mapped back to the 5C primer pseudo-genome using Bowtie v0.12.7.

Counts of primer-primer interactions were analyzed using previously published tools[75]. Briefly, outlier counts (which arise from 5C primer biases) were removed if their value was 8 fold higher than the median of a $5 \times 5$ primer-primer count window centered on the location in the counts matrix of the primer-primer count in question. Because in the double alternating design up to 2 primers could map to a single restriction enzyme fragment, we next converted primer-primer counts to fragment-fragment counts by calculating the arithmetic mean of the primer counts that mapped to each fragment-fragment pair. Each 5C region was then evenly divided into 4 kb bins and fragment counts were 'binned' by summing the fragment-fragment counts that fell within a 12 kb × 12 kb window centered at the middle of the genomic coordinates of each bin-bin pair[76]. Binned count matrices were balanced using the Joint Express algorithm[75] and quantile-normalized for consistent condition-to-condition comparison. Balanced counts were normalized for background contact domain signal, which was modeled using the donut background filter[61,77]. The resulting background-normalized interaction frequency counts were fit with a logistic distribution from which p-values were computed for each bin-bin pair and converted into 'Background-corrected Interaction Scores' (interaction score = $-10*\log2(p\text{-value})$). Interaction scores have proven to be informatively comparable across replicates and conditions and as such were used for visualization analyses.

**Chromatin immunoprecipitation.** Chromatin immunoprecipitation (ChIP) assays were performed as previously described[78]. Cortical nuclei were isolated from frozen frontal cortices of mice with Nuclei EZ prep kit (NUC101-1KT, Sigma). Chromatin from ~100,000 nuclei were used for each ChIP reaction. Nuclei were fixed with 1% formaldehyde for 15 min and then immediately quenched by adding glycine stock solution to a final concentration of 125 mM for 5 min. Nuclei were then rinsed with cold PBS twice, incubated with ChIP lysis buffer (50 mM HEPES-KOH pH7.5, 140 mM NaCl, 1 mM EDTA pH 8, 1% Triton X-100, 0.1% sodium deoxycholate, 0.1% SDS, Protease Inhibitors) for 10 min on ice. Nuclei lysates were sonicated for

15 min using CovarisTM S220 (Covaris, Inc) to shear chromatin to an average fragment size of 300–900 bp. The following antibodies were used in ChIP: anti-YY1(#61779, Active Motif) and IgG (#2729 S, CST). After washing with low salt wash buffer (0.1% SDS, 1% Triton X-100, 2 mM EDTA, 20 mM Tris-HCl pH 8.0, 150 mM NaCl), high salt wash buffer (0.1% SDS, 1% Triton X-100, 2 mM EDTA, 20 mM Tris-HCl pH 8.0, 500 mM NaCl) and LiCl wash buffer (0.25 M LiCl, 1% IGEPAL, 1% sodium deoxycholate, 1 mM EDTA, 10 mM Tris-HCl pH 8.0) successively, DNA was eluted wiht elution buffer (1% SDS, 100 mM NaHCO3) and purified using QIAquick PCR purification kit (28104, Qiagen). Input and bound DNAs were amplified by quantitative real time PCR using the same QuantStudio 7 Flex platform. The enrichment of the bound DNA was calculated as a percentage of the input. Amplification primer sequences were designed against the Nr3c1 promoter: 5′ CACGGTCACCATTTTGGCAG-3′ and 5′- GTCTTACAGCCACGGCC TAC-3′; and Yy1 promoter: 5′-AAGAAGTGGGAGCAGAAGCA-3′ and 5′-ATGG CTTCCCTTCAAAACAA-3′, where YY1 was shown with significant enrichment binding as previously reported.

**Statistics.** Values are expressed as mean ± s.e.m. as indicated. Statistical analyses were performed using the GraphPad Prism 8 software and R version 3.5.2. Statistical tests and R packages used in this study are described in the figure legends and "Methods" section.

**Reporting summary.** Further information on research design is available in the Nature Research Reporting Summary linked to this article.

## Data availability

Sequencing data can be found in NCBI GEO with the accession code GSE145970. Source data are provided with this paper containing the raw data used in the charts/graphs in Figs. 1, 4, 5, 6, and Supplementary Figs. 4–7. Common scripts used for bulk nuclei RNA-seq, sNucDrop-Seq, and 5C analysis were described in previous studies, respectively[36,31,77]. All other data or resources are available from the corresponding author upon request. Source data are provided with this paper.

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

## Acknowledgements
We thank Dr. Michael Atchison for kindly providing *Yy1*$^{fl/fl}$ mice and acknowledge Sixing Chen for sharing his expertise in PFC sectioning, Yugong Hu for his assistance with the RNAScope and ChIP experiments, and Dasha Zaitseva, George Gardner, and Zhou Zhou for their assistance with behavioral testing. The Next-Generation Sequencing Core and the Penn Vector Core at the Perelman School of Medicine provided sequencing and AAV production services. This work is partially supported by the Brain Research Foundation (Z.Z.), NIH R01MH111719 and R01NS081054 (Z.Z.), R01NS114226 (J.E.P.-C.). D.Y.K. is an Alavi-Dabiri postdoctoral fellow and supported by the T32 Training Program in Neurodevelopmental Disabilities (T32NS007413). Z.Z. is a Pew Scholar in biomedical science.

## Author contributions
D.Y.K. and Z.Z. conceived and designed study; D.Y.K. performed experiments and collected data with assistance from B.X., J.H.N., and Y.C.; B.X. conducted RNAscope and ChIP-qPCR; P.H. and D.K. conducted sNucDrop-seq, P.H. analyzed sNucDrop-seq data under the guidance of H.W.; Y-T.Z. analyzed bulk RNA-seq data; J.B. performed 5C experiments and analyzed data under the guidance of J.E.P.-C.; J.A.B. guided CUS experiments; D.Y.K. wrote the original manuscript with input from co-authors and revised manuscript with Z.Z. Z.Z. acquired funding.

## Competing interests
The authors declare no competing interests.

## Additional information

**Peer review information** *Nature Communications* thanks Zhen Yan and the other anonymous reviewer(s) for their contribution to the peer review this work. Peer reviewer reports are available.

