## [Peer Review File · Nature Communications]

Reviewers' Comments:

Reviewer #2:

Remarks to the Author:

The authors addressed one of my major concerns, which was the validity/robustness of identifying the correct cell type using sNucDrop-seq.

They did not address another major concern. I understand there are limitations in their ability to perform immunohistochemistry. They could try other methods, such as single cell FISH or RNAscope. Given the authors cannot perform a pseudobulk analysis to corroborate the validity of their single cell analysis, I think they still need to use an orthogonal method to support their sNucDrop-seq findings.

They provide reasonable explanations for my other major concerns.

Reviewer #3:

Remarks to the Author:

The paper by Kwon et al has included transcriptome profiling (bulk nuclei and single-nucleus) showing transcriptional changes in cortical cells from animals subjected to chronic unpredictable stress (CUS). There are a few main findings: (1) CUS alters gene transcription and chromatin folding; (2) neocortical excitatory neurons are particularly vulnerable to CUS, and Ex_L2/3 neurons have the most changed genes; (3) Yin Yang 1 (YY1), a transcription factor involved in chromatin structure organization, is reduced by CUS; (4) YY1 ablation in PFC excitatory neurons enhances stress sensitivity following an abbreviated stress (aCUS) exposure; (5) loss of YY1 in PFC excitatory neurons provokes maladaptive behaviors in stressed females and males.

The study has used cutting-edge genomic and epigenomic approaches to identify the broad and cell type-specific transcriptional changes by CUS. In addition, it has attempted to find the causal gene mediating the behavioral and transcriptional effects of CUS. While the quantity of data and analyses is impressive, some conclusions are not compelling. One major concern is that despite the selection of YY1 as a focal candidate from the analysis of sequencing data of CUS mice, follow-up studies indicate that YY1 is NOT a causal gene for the behavioral effects of CUS. It also lacks direct evidence showing that YY1 leads to transcriptional dysregulation of key genes in CUS mice.

It seems that this paper has made little changes to address previous reviewers' comments on the submission to Nature Neuroscience. All the 6 figures are almost the same as before. A few specific concerns need to be addressed.

1. Fig. 1 does not provide new information, because many prior publications have shown CUS-related behavioral data, and it should be moved to supplementary figures. Some behavioral measurements are highly arbitrary and the interpretation is problematic. For example, Fig. 1f, coat state score and Fig. 1g, nest score. Why not perform the direct measurement of grooming behavior? Since "nests between control and CUS animals were virtually indistinguishable after 24 hours (data not shown)" (p6), the small difference at 16-hour in Fig. 1g does not have too much meaning. These behavioral assays should be deleted.
2. The description of Fig. 2d is not very consistent with the GO terms of DEGs, for example, "membrane-bound receptors for different classes of neurotransmitters" can not be found on the GO terms.
3. Fig. 2h and 2i have redundant presentation, which is unnecessary. For Fig. 2i, what is the

functional consequence of the increased chromatin interactions of YY1 and CTCF motif sequences around Syt1 locus in CUS-subjected mice? Is there any difference in H3K27ac or Syt1 expression between Control vs. CUS?

4. Fig. 2g shows a negative enrichment of primary response genes, consistent with the down-regulation of activity-dependent genes, such as Fos, Fosb, Fosl2 (p8). However, Fig. 4c shows the upregulation of such genes, including Fos, Fosb, Egr1/2/4, in layer 2/3 excitatory neurons. How to interpret the opposing results? Many studies have shown that CUS leads to decreased neuronal activity in the PFC, which is correlated with the decreased expression of activity-dependent genes.

5. Fig. 4a and 4b have redundant components. Fig. 4a is not needed.

6. Fig. 4c, YY1 is one of the down-regulated genes at the bottom of the list, with a very small change (close to 0). It is not convincing to focus on YY1 instead of other TFs that show more significant changes. YY1 is unlikely to be the key causal gene mediating CUS effects. In agreement with this, in vivo deletion of Yy1 in PFC excitatory neurons fails to induce depressive- and anxiety-like behaviors like CUS (Supplemental Fig 6).

7. Fig. 5l and 5m show the enrichment of YY1 binding at promoter/enhancer regions of Nr3c1 and Fos genes, but there is no direct evidence showing differences on YY1 signals between Control and CUS. It also lacks evidence showing that YY1 leads to transcriptional dysregulation of these genes.

8. Fig. 5j and 5k show the reduction of Fos and Fosl2 in YY1-exKO and CUS, which is also opposite to the "Up" of these genes shown in Fig. 4c. How to explain this?

9. Fig. 6 shows the data in females, but almost all the results are the same as those from males. The only minor difference is Fig. 6n and 6o, which show marginal significance on two measurements (Sucrose Preference and Nesting) by YY1 exKO. Fig. 6n (female) and Fig. 5h (male) have very similar results, except that control females somehow have smaller variations. Fig. 6o (female) and Fig. 5i (male) are also largely the same, and this behavioral assay is even problematic, and should be removed (see above). Given the high similarity of data between males and females, Fig. 6 should be moved to supplementary figures.

Reviewer #4:

Remarks to the Author:

Comments to the authors

The authors did an excellent job addressing most of the comments from Reviewer1 and Reviewer4. The authors could address the remaining concern listed below.

1. Down-sampling of sNucDrop-seq data suggests that the number of DEGs in Ex_L2/3_Enpp2 is still higher than other cell types. However, other cell types such as Ex_L5 and Ex_L2/3_Ndst4 now show comparable DEGs to those of Ex_L2/3_Enpp2. I am afraid that the result could still be a chance of the particular 100 cells selected. The author could repeat the random downsampling, call DEG multiple times, and test if the distribution of DEG numbers differ among the cell type.

2. I appreciate that the author provided input and control IgG track. I am still not convinced that the YY1 signal at the Fos locus is genuine. To me, it looks like the sequence depth of input and IgG samples are much shallower than YY1 ChIP-seq. The Y-axes of the tracks are not labeled. I would suggest to combine the reads from Input rep1, rep2, IgG, rep1, and rep2 and compare signals with comparable sequencing depths. Was any peak caller able to identify that YY1 signals as a peak? YY1 signals of Nr3c1 and YY loci look good.

I have some concerns with the below comments reviewer 1 made, and I agree with the author's responses.

- The story would be greatly improved in the bulk RNA seq data were removed completely and focus on the really nice cell type specific effects of CUS using single cell. Having said this, it would

essential to extend this analysis to females.

- How are IEGs detected at baseline and then shown as reduced? By their very nature, IEG expression should be near zero in the home cage.
- The use of cell culture to infer what is happening in the adult brain is difficult to embrace as a reflection of neural activity associated with CUS. It may be best to keep the analysis primarily in vivo and move all of the in vitro work into supplemental.
- Perhaps by directing YY1 to enhance the expression of key synapse related genes that are also involved in stress resilience, a reversal of the effect of CUS could be shown and would thus be a powerful demonstration of the important role of YY1 in stress regulation.

Point-by-point Response to Reviewers' Comments

Reviewer #2 (Remarks to the Author):

The authors addressed one of my major concerns, which was the validity/robustness of identifying the correct cell type using sNucDrop-seq.

We thank this reviewer for their comment.

They did not address another major concern. I understand there are limitations in their ability to perform immunohistochemistry. They could try other methods, such as single cell FISH or RNAscope. Given the authors cannot perform a pseudobulk analysis to corroborate the validity of their single cell analysis, I think they still need to use an orthogonal method to support their sNucDrop-seq findings.

We understand the reviewer's concern. We have now subjected independent cohorts of mice to CUS and performed RNAscope using specific probes designed against *Yy1*, a differentially expressed gene (DEG) upon CUS, as well as *Cdkl5*, which is not altered by CUS, in medial prefrontal cortical sections from the brains of control and CUS mice as the reviewer requested (see Figure 1 below). We quantified and compared the numbers of *Yy1* and *Cdkl5* mRNA puncta in layer 2/3 cells to those captured in layer 6. In agreement with our sNucDrop-seq results, we found that levels of *Yy1* mRNA puncta were significantly decreased selectively in layer 2/3 in the medial PFCs of CUS mice (see below Fig 1; L2/3, $P=0.0198$; L6, $P=0.276$). This finding was in contrast to *Cdkl5*, which we found was neither altered in layer 2/3 nor in layer 6 (Fig 1 below; L2/3, $P=0.466$; L6, $P=0.833$). These data are now in the revised manuscript as Fig 4f and 4g.

Figure 1. Representative RNAscope images for *Yy1* and *Cdkl5* in medial PFC tissues taken from control and CUS mice. Quantification of *Yy1* and *Cdkl5* mRNA puncta number per cell in

*layer 2/3 and layer 6 are shown on the right (Linear mixed-effects analysis; n=30 cells per animal, n=3 animals per condition). *P< 0.05.*

We believe that these findings, in addition to the consistent changes in YY1 expression and regulon activity across two separate CUS cohorts and sNucDrop-seq results (Fig 4 and Supplemental Fig 3), lend strong support to our findings from our single cell data. We thank the reviewer for suggesting this validation experiment.

They provide reasonable explanations for my other major concerns.

We thank this reviewer for their valuable comments and suggestions, which we believe has strengthened our study.

Reviewer #3 (Remarks to the Author):

The paper by Kwon et al has included transcriptome profiling (bulk nuclei and single-nucleus) showing transcriptional changes in cortical cells from animals subjected to chronic unpredictable stress (CUS). There are a few main findings: (1) CUS alters gene transcription and chromatin folding; (2) neocortical excitatory neurons are particularly vulnerable to CUS, and Ex_L2/3 neurons have the most changed genes; (3) Yin Yang 1 (YY1), a transcription factor involved in chromatin structure organization, is reduced by CUS; (4) YY1 ablation in PFC excitatory neurons enhances stress sensitivity following an abbreviated stress (aCUS) exposure; (5) loss of YY1 in PFC excitatory neurons provokes maladaptive behaviors in stressed females and males.

The study has used cutting-edge genomic and epigenomic approaches to identify the broad and cell type-specific transcriptional changes by CUS. In addition, it has attempted to find the causal gene mediating the behavioral and transcriptional effects of CUS. While the quantity of data and analyses is impressive, some conclusions are not compelling. One major concern is that despite the selection of YY1 as a focal candidate from the analysis of sequencing data of CUS mice, follow-up studies indicate that YY1 is NOT a causal gene for the behavioral effects of CUS. It also lacks direct evidence showing that YY1 leads to transcriptional dysregulation of key genes in CUS mice.

It seems that this paper has made little changes to address previous reviewers' comments on the submission to Nature Neuroscience. All the 6 figures are almost the same as before. A few specific concerns need to be addressed.

We thank the reviewer for re-reading our manuscript. In this round of revision, we have specifically addressed the major concern regarding the causal role of YY1 in mediating stress responses, provided new direct evidence showing CUS-induced YY1 reduction leads to down-regulation of YY1 targets, and also highlighted all major changes and new data for this reviewer's convenience.

1. Fig. 1 does not provide new information, because many prior publications have shown CUS-related behavioral data, and it should be moved to supplementary figures. Some behavioral measurements are highly arbitrary and the interpretation is problematic. For example, Fig. 1f,

coat state score and Fig. 1g, nest score. Why not perform the direct measurement of grooming behavior? Since “nests between control and CUS animals were virtually indistinguishable after 24 hours (data not shown)” (p6), the small difference at 16-hour in Fig. 1g does not have too much meaning. These behavioral assays should be deleted.

We thank this reviewer for providing critical input into the presentation and assessment of behavioral phenotypes. There are several CUS paradigms reported in literature; the one we employed in this study is a 12-day stress paradigm that was developed by our collaborator, Dr. Julie Blendy (which itself was adapted from a paradigm developed by the Duman lab at Yale). The advantage of this paradigm is that it, unlike most other chronic stress paradigms, drives depressive- and anxiety-like behaviors after only 12 days of stress (mice are stressed 3x a day, including overnight; many other chronic stress paradigms subject mice to stressors once or twice a day and require several weeks of stress to induce behavioral phenotypes). While Dr. Blendy has published work using this paradigm in mice (*Yohn et al., Neuropsychopharmacology, 2017*), the behaviors in her published study were only measured in adult mice **30 days after CUS** was delivered. Importantly, there is **minimal** overlap between the behaviors she tested in her study and ours—the only behavioral test in common between the two is the elevated zero maze, and their study did not find any CUS-induced changes in EZM behavior 30 days after CUS. There is also minimal overlap between the behavioral tests our study and the one that the Duman lab published using CUS. Thus, we believe the data we collected and presented here, measuring the effects of CUS in a **comprehensive** manner in mice, are largely novel and different from previously reported studies.

In Fig 1f, we assayed the deterioration of coat state in our mice as an indirect measure of grooming because it is a **quick and minimally stressful procedure** in which an animal is gently handled by the experimenter for visual inspection. To directly measure coat state, an animal would need to be singly housed in a chamber connected to a camera and video recorded for an extended period of time to capture enough spontaneous grooming bouts. This alone introduces stress to the animal (e.g., social isolation in a novel environment) and this stress could confound grooming behavior in control animals. The nesting assay (and related nestlet shredding assays) as shown in Fig 1g that we performed in this study has been reported in other studies as a motivated behavior and disrupted by chronic stress (ex. *Manners et al., Brain Behav Immun, 2019*). Our data shows that CUS animals are **slower** to build nests when placed in a new cage with a fresh cardboard nestlet, but **not physically incapable** of building nests (as shown by the difference in nesting between 16 and 24 hours), which suggests that CUS decreases motivated nesting behavior. For these reasons we believe these data provide an important picture of behavioral maladaptation in mice to CUS exposure (page 6). We are happy to move these data to the Supplemental Figures if this reviewer insists.

2. The description of Fig. 2d is not very consistent with the GO terms of DEGs, for example, “membrane-bound receptors for different classes of neurotransmitters” can not be found on the GO terms.

We thank the reviewer for pointing this out. We drew this conclusion based on the collective information from the top GO terms, including “plasma membrane”, “cell periphery”, “plasma membrane bounded cell projection”, etc, which include genes encoding membrane-bound receptors for different neurotransmitters, but we have revised this sentence in the manuscript to make it consistent (page 8).

3. Fig. 2h and 2i have redundant presentation, which is unnecessary. For Fig. 2i, what is the functional consequence of the increased chromatin interactions of YY1 and CTCF motif sequences around *Syt1* locus in CUS-subjected mice? Is there any difference in H3K27ac or *Syt1* expression between Control vs. CUS?

We appreciate this reviewer for bringing up this point. Fig 2h depicts a birds-eye view of all of the chromatin-chromatin interactions at the **entire genomic locus** encompassed by the 5C primer set we used, as well as the location of a zoomed-in inset focusing on the genomic region upstream of the *Syt1* transcriptional start site (TSS) where chromatin contacts are altered in CUS samples compared to controls. Fig 2i illustrates additional layers of genomic information at this locus, including CTCF and YY1 binding motifs, H3K27ac ChIP-seq tracks from the mouse cortex, and H3K27ac ChIP-seq tracks from cultured cortical neurons upon Bicuculline or TTX treatment at the same upstream region of *Syt1* with sites of increased chromatin contact. While both figure panels show the same genomic locus, the data are not redundant and combining the two would make it difficult to visually discriminate all of the different layers of genomic information we present.

Like the reviewer we also asked what the functional consequence of increased chromatin interactions at the YY1 and CTCF motif sequences are and have carefully examined *Syt1* expression in control and CUS mice in our RNA-seq datasets. *Syt1* is not significantly altered in CUS mice in either our bulk or single nucleus sequencing datasets. This does not exclude the possibility that different isoforms of *Syt1* are altered by CUS, as nuclear RNA-seq data limits our ability to identify splicing isoforms. However, we are impressed by the **nearly identical** changes in chromatin contacts induced by CUS and by stimulating neuronal activity via Bicuculline, in contrast to silencing neuronal activity via TTX (Supplemental Figure 1). We decide to keep this 5C data in the revision, but have significantly revised our manuscript, from the Title, Abstract, Main Text to Discussion, and toned down the implication of those changes in chromatin contacts upon CUS. We believe locus-specific manipulation of chromatin-chromatin interactions, coupled with comprehensive profiling of YY1-associated chromatin contacts in CUS versus control conditions, would be needed to gain insights into the functional consequences of altered chromatin contacts and that is likely beyond the scope of this current study. Our presented data merely implies CUS shapes the PFC into a state of reduced neuronal activity by decreasing the transcription of neuronal activity-dependent genes and restructuring high-order genome architecture into a pattern associated with synaptic silencing (Fig 2 and Supplemental Fig 1) (page 9).

4. Fig. 2g shows a negative enrichment of primary response genes, consistent with the down-regulation of activity-dependent genes, such as *Fos*, *Fosb*, *Fosl2* (p8). However, Fig. 4c shows the upregulation of such genes, including *Fos*, *Fosb*, *Egr1/2/4*, in layer 2/3 excitatory neurons. How to interpret the opposing results? Many studies have shown that CUS leads to decreased neuronal activity in the PFC, which is correlated with the decreased expression of activity-dependent genes.

We regret that our explanation of SCENIC analysis in the last round of rebuttal did not reach an agreement with this reviewer. The figure in 4c generated by the **SCENIC** analysis **does not** represent dysregulation of transcription factor (TF) gene expression (only 5 of these were significantly deregulated by single nucleus RNA-seq and these are marked with red asterisks in Fig 4c), but rather depicts deregulation of the **gene regulatory network or Regulon**. SCENIC is used for single cell gene regulatory network analysis and reconstructs regulons, which are transcription factors and their target genes. TFs for each regulon are determined by RcisTarget,

which identifies over-represented TF binding motifs in a gene-set. The activity of each regulon is quantified by the regulon's target genes (AUcell), **not** the transcript level of the TF itself. So, the fact that the regulons for Fos and Fosb increased while their expression itself decreased in our bulk nuclei RNA-seq dataset suggests that Fos and Fosb regulatory activity increased in cortical excitatory neurons during CUS. We envision that increased regulatory activity may have occurred earlier in the CUS paradigm, as acute stress is known to activate mPFC glutamatergic neurons. But, as evidenced by their gene expression levels, the transcription of these genes themselves had already decreased by the time we harvested PFC tissues for nuclear RNA-seq, after the 12th day of CUS. As the reviewer mentioned, chronic stress is associated with decreased neural activity in the PFC, which are **consistent** with our findings of decreased expression of activity dependent genes in the PFC, including Fos, Fosb, and Fosl2, and our 5C data showing a pattern of chromatin interactions associated with neuronal silencing. We understand that this can be confusing, and we thank this reviewer for raising this question. We have now clarified this point and added a new paragraph in the Discussion section of our manuscript (pp. 26-27, highlighted) in case other readers also misinterpret Fig 4c (page 20).

5. Fig. 4a and 4b have redundant components. Fig. 4a is not needed.

We thank the reviewer for this comment but include figure 4a and 4b for specific reasons. Fig 4a is a GO analysis of all DEGs identified via bulk nuclei RNA-seq, and Fig 4b shows GO terms for upregulated DEGs and downregulated DEGs separately. These are not the same analyses; downregulated DEGs happened to enrich the same GO terms as those found from all DEGs, which is the point we made in our manuscript. If this reviewer insists, we are happy to move Fig. 4a or 4b into supplemental figures (page 13).

6. Fig. 4c, YY1 is one of the down-regulated genes at the bottom of the list, with a very small change (close to 0). It is not convincing to focus on YY1 instead of other TFs that show more significant changes. YY1 is unlikely to be the key causal gene mediating CUS effects. In agreement with this, in vivo deletion of Yy1 in PFC excitatory neurons fails to induce depressive- and anxiety-like behaviors like CUS (Supplemental Fig 6).

We thank this reviewer for pointing this out and regret that we did not make our points clear in the last round of rebuttal and revision. As mentioned in the above explanation to comments #4, Fig 4c illustrates findings via **SCENIC analysis, which represent changes in gene regulatory network, but not necessarily in the expression level of the representative TF**. Our sNucDrop-seq data show that Yy1 transcript levels decrease ~30% (Fig 4d), and new RNAscope experiments we performed also show decreased Yy1 mRNA levels of ~45% in the mPFC by RNAscope (Fig 4f and 4g) (page 14).

The above data clearly support the involvement of YY1 in CUS adaptation. To determine a potential causal or functional role of YY1 in stress-related behavioral maladaptation, we first carried out cell culture experiment of primary neurons, followed by exposure to CORT, and confirmed that both YY1 mRNA and protein expression are down-regulated upon CORT exposure (Supplemental Figure 5). We then took an AAV-mediated genetic approach and selectively decreased Yy1 expression in vivo in mPFC excitatory neurons in both male and female mice. Notably, we found that selective reduction of Yy1 expression in mPFC excitatory neurons does not drive behavioral changes in naïve mice but rather reduces their ability to cope with stress when subjected to a short 3-day aCUS paradigm (Figures 5 and 6, Supplementary Figure 6). Based on these data, we do not argue that ablation of YY1 alone would induce

anxiogenic and depressive-like behaviors—rather our findings demonstrate that Yy1 plays a critical role in **stress adaptation**. We believe this is a rather important and novel finding to the field, and implicate YY1, a well-studied transcriptional factor, in stress response for the first time. We have now revised our manuscript to describe the rational why focusing YY1 in several sections of the manuscript (see highlighted paragraphs).

7. Fig. 5l and 5m show the enrichment of YY1 binding at promoter/enhancer regions of Nr3c1 and Fos genes, but there is no direct evidence showing differences on YY1 signals between Control and CUS. It also lacks evidence showing that YY1 leads to transcriptional dysregulation of these genes.

We thank this reviewer for pointing out this concern. To specifically address these questions, we carried out an independent CUS treatment of mice, followed by chromatin immunoprecipitation using an antibody against YY1 (ChIP). We focused our ChIP study on frontal cortical tissues from control and CUS mice and specifically measured YY1 enrichment at two genomic regions that show selective binding of YY1 (Figure 5l and 5m) with corresponding genes showing reduction of expression in CUS condition in comparison to controls (Figure 5k). Notably, we found that the binding of YY1 at the promoter regions of Nr3c1 and Yy1 is significantly decreased in CUS tissues, compared to controls, suggesting that reduced binding of YY1 decreases transcription of *Nr3c1*, a likely direct target of YY1, and *Yy1* itself via auto-regulation. We believe these data provide the first direct evidence linking YY1 to target gene regulation, in the context of stress exposure, and have now included this new data in our revised manuscript as Fig 5n and 5o, shown as figure 2 below (page 21-22).

Figure 2. Levels of YY1 ChIP-qPCR signal at the *Yy1* promoter decrease in frontal cortices of CUS mice relative to controls (Mann-Whitney U-test; $P= 0.028$; $n=4$ per group). (o) Levels of YY1 ChIP-qPCR signal at the *Nr3c1* promoter decrease in frontal cortices of CUS mice relative to controls (Unpaired t-test with Welch's correction; $P= 0.029$; $n=4$ mice per group).

8. Fig. 5j and 5k show the reduction of Fos and Fosl2 in YY1-exKO and CUS, which is also opposite to the “Up” of these genes shown in Fig. 4c. How to explain this?

Please see our response to Comments #4 above. The reduction of Fos and Fosl2 expression in our RNA-seq data is consistent with reduced neuronal activity and decreased neuronal activity-dependent gene expression in CUS conditions. The “Up” in Fig 4c refers to Fos and Fosl2 Gene Regulatory Network (GRN) or Regulon. We have now added a few annotations to avoid potential confusion to readers.

9. Fig. 6 shows the data in females, but almost all the results are the same as those from males. The only minor difference is Fig. 6n and 6o, which show marginal significance on two measurements (Sucrose Preference and Nesting) by YY1 exKO. Fig. 6n (female) and Fig. 5h (male) have very similar results, except that control females somehow have smaller variations. Fig. 6o (female) and Fig. 5i (male) are also largely the same, and this behavioral assay is even problematic, and should be removed (see above). Given the high similarity of data between males and females, Fig. 6 should be moved to supplementary figures.

We thank the reviewer for this comment. We thought that replicating CUS studies in female mice and replicating functional assessment of YY1 in stress adaptation via AAV-mediated reduction in female mice represent a strength in our study, as most of previous similar studies have focused on male animal models, despite stress-related major depressive disorder occurs more frequently in women than in men. Despite of similar results, the differences between males and females are particularly interesting in our view. Thus, we incline to include our findings in female mice in the main figure but are happy to re-consider if this reviewer insists.

Reviewer #4 (Remarks to the Author):

The authors did an excellent job addressing most of the comments from Reviewer1 and Reviewer4. The authors could address the remaining concern listed below.

We thank the reviewer for their time in reviewing our responses to both their concerns as well as Reviewer 1's.

1. Down-sampling of sNucDrop-seq data suggests that the number of DEGs in Ex_L2/3_Enpp2 is still higher than other cell types. However, other cell types such as Ex_L5 and Ex_L2/3_Ndst4 now show comparable DEGs to those of Ex_L2/3_Enpp2. I am afraid that the result could still be a chance of the particular 100 cells selected. The author could repeat the random down-sampling, call DEG multiple times, and test if the distribution of DEG numbers differ among the cell type.

We thank the reviewer for this suggestion. As requested, we repeated the random down sampling 30 times and summarized proportion of DEG numbers in each cell type (shown as Fig 3 below). Ex_L2/3_Enpp2 cluster continues to demonstrate the highest the number of DEGs (mean: 0.517%, median: 0.497%) among all the clusters. Compared to the 2nd highest Ex_L4 cluster, Ex_L2/3_Enpp2 cluster shows a significantly higher proportion of DEG numbers (two-sided Wilcox test, $P = 6.26e-05$). We have now added this data into a new panel in Supplemental Figure 2a) (page 12).

Figure 3. Boxplot showing the proportion of DEG numbers in each cell type after 30 times of repeated down-sampling.

2. I appreciate that the author provided input and control IgG track. I am still not convinced that the YY1 signal at the Fos locus is genuine. To me, it looks like the sequence depth of input and IgG samples are much shallower than YY1 ChIP-seq. The Y-axes of the tracks are not labeled. I would suggest to combine the reads from Input rep1, rep2, IgG, rep1, and rep2 and compare signals with comparable sequencing depths. Was any peak caller able to identify that YY1 signals as a peak? YY1 signals of Nr3c1 and YY loci look good.

This is an excellent suggestion and indeed the sequencing depth of the input and IgG samples obscured comparison. We have since removed the Fos figure panel and replaced it with YY1 ChIP-seq tracks at the *Yy1* genomic locus (Fig 5I). We also validated that the YY1 signal at *Yy1* and *Nr3c1* was indeed called as peaks by MACS2. Furthermore, we performed independent ChIP qPCR for YY1 at both the *Yy1* and *Nr3c1* promoters, using primers designed against the sequences where YY1 peaks were called, and found an enrichment of YY1 ChIP signal at these sites by qPCR using an independent cohort of CUS and control mice. Importantly, YY1 binding at both of these promoters was decreased in CUS mice, indicating decreased regulation of these genes by YY1 with chronic stress (new Fig 5n,o, and shown as figure 2 above).

I have some concerns with the below comments reviewer 1 made, and I agree with the author's responses.

- The story would be greatly improved in the bulk RNA seq data were removed completely and focus on the really nice cell type specific effects of CUS using single cell. Having said this, it would essential to extend this analysis to females.
- How are IEGs detected at baseline and then shown as reduced? By their very nature, IEG expression should be near zero in the home cage.
- The use of cell culture to infer what is happening in the adult brain is difficult to embrace as a reflection of neural activity associated with CUS. It may be best to keep the analysis primarily in vivo and move all of the in vitro work into supplemental.
- Perhaps by directing YY1 to enhance the expression of key synapse related genes that are also involved in stress resilience, a reversal of the effect of CUS could be shown and would thus be a powerful demonstration of the important role of YY1 in stress regulation.

We greatly appreciate Reviewer 4 for taking their time going over Reviewer 1's comments and making valuable recommendations strengthening our manuscript.

Reviewers' Comments:

Reviewer #2:

Remarks to the Author:

The authors have addressed my remaining concerns.

Reviewer #3:

Remarks to the Author:

The authors have significantly improved the revised paper by addressing previous concerns with new data and explanations. Before its publication, there are a few minor concerns that need to be addressed.

1. in vivo deletion of Yy1 in PFC excitatory neurons fails to induce depressive- and anxiety-like behaviors. The statement in Abstract "Selective ablation of YY1 in neocortical excitatory neurons enhances stress sensitivity in male and female mice, inducing depressive- and anxiety-related behaviors..." is a bit confusing and needs to be corrected.

2. Fig. 2h and 2i should be consolidated to avoid the redundant presentation of the same components.

3. Fig. 6 repeats the same set of data in females as that from males. Given the high similarity and the lack of a focus on sex differences in this paper, it should be moved to the section of supplementary figures.

4. Some references in the newly added texts (highlighted) are not incorporated in the bibliography.

Reviewer #4:

Remarks to the Author:

The authors addressed the reviewer's concerns well in most cases. Now the work is better substantiated with stronger evidence, thus provides unique insights into how YY1 contributes to stress adaptation in mice. I appreciate the authors' efforts and believe that the work merits publication in Nature Communications.

REVIEWERS' COMMENTS

Reviewer #2 (Remarks to the Author):

The authors have addressed my remaining concerns.

Thank you very much.

Reviewer #3 (Remarks to the Author):

The authors have significantly improved the revised paper by addressing previous concerns with new data and explanations. Before its publication, there are a few minor concerns that need to be addressed.

1. in vivo deletion of Yy1 in PFC excitatory neurons fails to induce depressive- and anxiety-like behaviors. The statement in Abstract “Selective ablation of YY1 in neocortical excitatory neurons enhances stress sensitivity in male and female mice, inducing depressive- and anxiety-related behaviors...” is a bit confusing and needs to be corrected.

We wrote in our abstract that “Selective ablation of YY1 in neocortical excitatory neurons enhances stress sensitivity in male and female mice inducing depressive- and anxiety-related behaviors... **following an abbreviated stress exposure**” to describe that the maladaptive behaviors are induced in these animals due to an enhanced stress sensitivity from YY1 deletion in neocortical excitatory neurons, which our study shows. We agree that this statement may be confusing and so have rewritten it to say, “Selective ablation of YY1 in cortical excitatory neurons enhances stress sensitivity in male and female mice and alters the expression of stress-associated genes following an abbreviated stress exposure”, exactly as revised by the editor.

2. Fig. 2h and 2i should be consolidated to avoid the redundant presentation of the same components.

We have consolidated Figs 2h and 2i into one panel as the reviewer requested.

3. Fig. 6 repeats the same set of data in females as that from males. Given the high similarity and the lack of a focus on sex differences in this paper, it should be moved to the section of supplementary figures.

We appreciate the reviewer's suggestion. However, it is exactly because we see similar results in male and female mice that we would like to keep figure 6 in the main body of the manuscript. The fact that we were able to replicate the function of YY1 in stress adaptation in both male and female mice is evidence of the central role this protein plays in stress response, which is independent of many of the endocrine or sex-specific behavior effects normally associated with females. It is our opinion that this data would be of interest to other scientists in the field and thus warrants it to be a main figure. In addition, as this reviewer is likely aware, most of previous stress-related studies have focused on male animal models, despite stress-related major depressive disorder occurs more frequently in women than in men. Thus far, there has been very few studies reporting the effect of stress in both sexes of mice in a single manuscript.

Despite of similar results, the few noted differences between males and females, however, are particularly interesting in our view (see our discussion section on this point). Thus, we would like to ask the editor to make the final call.

4. Some references in the newly added texts (highlighted) are not incorporated in the bibliography.

We thank the reviewer for pointing this out and we have incorporated these references into our bibliography.

Reviewer #4 (Remarks to the Author):

The authors addressed the reviewer's concerns well in most cases. Now the work is better substantiated with stronger evidence, thus provides unique insights into how YY1 contributes to stress adaptation in mice. I appreciate the authors' efforts and believe that the work merits publication in Nature Communications.

Thank you very much. This is rewarding to hear.